# Action Observation Training for Upper Limb Stroke Rehabilitation: A Pilot Study on the Role of Attention

**DOI:** 10.3390/jcm14186618

**Published:** 2025-09-19

**Authors:** Giada Milani, Andrea Baroni, Martina Galluccio, Giulia Fregna, Annibale Antonioni, Sofia Straudi, Thierry Pozzo, Luciano Fadiga

**Affiliations:** 1Department of Neuroscience and Rehabilitation, University of Ferrara, 44121 Ferrara, Italy; 2Iit@Unife Center for Translational Neurophysiology, Istituto Italiano Di Tecnologia, 44121 Ferrara, Italy; 3Department of Neuroscience, Ferrara University Hospital, 44124 Ferrara, Italy; 4INSERM U1093-Cognition, Action et Plasticité Sensorimotrice, Université de Bourgogne Franche-Comté, 21078 Dijon, France

**Keywords:** action observation training (AOT), attention, engagement, motor recovery, neurorehabilitation, stroke

## Abstract

**Background**: Restoring motor function is crucial for daily life after a stroke. Although patients’ engagement and attention influence motor recovery, these factors are frequently overlooked in rehabilitation interventions. **Methods**: This prospective open-label pilot trial (NCT04622189) investigated the impact of attentional deficits on engagement and motor recovery in 10 subacute stroke patients undergoing a 4-week action observation training program. At baseline, they were divided into two subgroups based on attentional performance, as determined by scores on the Test of Attentional Performance (subtests of divided attention and Go/No-Go): those with attention deficits (AD, i.e., deficits in one or both tasks, n = 6) and those without (No_AD, no deficits in either task, n = 4). **Results**: Both groups exhibited similar motor profiles at baseline; however, the AD group presented significantly lower cognitive reserve (AD mean (SD) 92.2 ± 4.09, No_AD 120 ± 14.9, *p* = 0.005) and greater anxiety and depressive symptoms (AD 66.7%, No_AD 0%, *p* = 0.035). While all patients showed improvements in motor outcomes, the No_AD group demonstrated significantly greater gains in upper limb function, as assessed by the Fugl-Meyer Assessment (AD 3.33 ± 1.21, No_AD 10.8 ± 5.7, *p* = 0.013). Engagement and accuracy of interactive questions, used as proxies for concentration during training, were also higher in the No_AD group and positively correlated (rho = 0.9075, *p* ≤ 0.001). Moreover, patients with attention deficits reported lower levels of engagement during training. **Conclusions**: These findings indicate that attentional status may affect both adherence to and responsiveness to rehabilitation. This highlights a potentially relevant factor to consider when improving post-stroke interventions.

## 1. Introduction

Stroke is a leading cause of disability in adults, with most patients experiencing upper limb motor dysfunction, cognitive impairments, and neuropsychiatric issues [1,2,3,4], which impact their quality of life and return to work [5,6]. Among cognitive issues, attention deficits are the most typical in stroke patients [7,8]. Indeed, they affect up to 46–92% of patients during the acute phase, with recovery paths that vary over the following weeks [9,10]. These impairments encompass a wide range, from decreased concentration and distractibility to limited multitasking ability, ultimately impairing daytime functioning and independence [11]. Importantly, beyond their immediate effects, attention deficits are consistently linked to negative outcomes, including a higher risk of falls in older adults and poorer long-term functional recovery [10,12]. This suggests that attentional dysfunction may act less as an isolated symptom and more as a determinant of rehabilitation potential. Indeed, difficulties in maintaining, dividing, or selectively focusing attention have been reported to correlate with decreased mobility, balance, and independence in daily activities up to one year after stroke [9]. Nonetheless, these associations were largely diminished after accounting for baseline functional status, indicating that attentional performance may have a contributory—albeit not primary—role in influencing long-term recovery outcomes, with baseline motor ability emerging as a more robust predictor. Overall, these findings highlight that attentional capacity is not only highly vulnerable after stroke but also plays a crucial role in shaping rehabilitation progress, as attentional status may be a modifiable factor influencing both adherence to and outcomes of rehabilitation.

Crucially, stroke rehabilitation is most effective when therapists encourage patients’ active participation and engagement [13,14,15]. However, this engagement appears to be affected by patients’ emotional state and cognitive functioning level [13,16]. Consistently, from a neuropsychological perspective, increased patient engagement seems to be associated with greater attention during exercise [17,18,19]. Furthermore, several studies have shown that positive clinical outcomes (i.e., reduced depression and better cognitive and motor outcomes) correlate with effective attention [13,20,21]. Taken together, these findings highlight the crucial role of attention as a mediator between patient engagement and rehabilitation outcomes, indicating that attentional capacity not only aids active participation during therapy but also significantly influences the course of recovery after stroke.

It is essential to acknowledge that patients with severe motor deficits often present with concomitant attentional impairments, as their lesions frequently extend to regions involved in both cognitive and motor functions [22,23]. Furthermore, their level of impairment generally hinders them from engaging in standard neurorehabilitation protocols that demand at least some active participation [24,25]. Since these patients generally have the poorest long-term outcomes, there is an urgent need for strategies applicable to individuals with severe motor impairments that can also address the cognitive dimensions of recovery, with attention being a particularly important target [26]. Recently, new rehabilitation strategies have been proposed to increase upper limb motor recovery after stroke, including action observation training (AOT), a mirror neuron system (MNS)-based approach that typically involves observing and performing goal-directed daily actions [27,28,29,30]. Specifically, it induces targeted motor facilitation in the corticospinal system, increasing the excitability of the injured sensorimotor system in the primary motor cortex and promoting brain reorganization by activating central representations of actions through the MNS [28]. It was also consistently demonstrated that AOT prevents corticomotor depression induced by immobilization during limb inactivity [31]. Importantly, since AOT depends on activating the MNS and related frontoparietal networks, it is closely connected with attentional control systems [32]. Therefore, successful engagement in AOT requires focusing attention to observe, process, and mentally simulate the watched motor actions [33]. Moreover, directing visual attention explicitly during AOT facilitates corticospinal excitability and may accelerate motor relearning through observation [34]. Attention deficits may limit the ability to properly encode and integrate observed movements, thereby reducing the effectiveness of AOT [35]. On the other hand, structured AOT protocols might also help train attention skills, as patients must repeatedly stay focused and selectively process relevant motor cues [36]. Thus, attention is both a necessary prerequisite and a potential focus in AOT, and understanding this relationship is essential for optimizing patient selection and therapeutic results.

Despite this body of evidence, attentional deficits have seldom been investigated as an explicit factor shaping the effectiveness of rehabilitation, particularly within structured interventions such as AOT. Indeed, most of the literature on AOT primarily focused on motor outcomes, often overlooking the dynamic contribution of attention to sustaining engagement and maximizing treatment efficacy [37,38,39]. Moreover, reliable in-treatment measures of attentional performance have rarely been incorporated, leaving the extent to which attentional deficits hinder participation and responsiveness largely unclear. Therefore, the present pilot study was designed to explore whether attentional performance influences both engagement and motor recovery during a four-week AOT program in subacute stroke patients. Specifically, we aimed to examine the interplay between attentional deficits, patients’ engagement, and their potential to benefit from AOT, while also determining whether AOT could foster improvements in attentional functioning. We hypothesized that attentional performance would significantly influence both adherence to and motor gains from AOT, and that, in turn, the structured demands of sustained concentration and interactive tasks in AOT might further boost participants’ attentional capacities.

## 2. Materials and Methods

### 2.1. Study Design

We conducted a prospective, open-label clinical trial from November 2020 to May 2022. Specifically, subacute stroke patients were recruited at Ferrara University Hospital (Italy). The subjects were evaluated and interviewed by a trained physiotherapist and psychologist before being included in the study, and their assessments determined whether they met the inclusion/exclusion criteria. All participants met the following inclusion criteria: (1) males or females aged 18 years or older; (2) diagnosis of first cerebral stroke (ischemic or hemorrhagic) verified by brain imaging within eight weeks; (3) upper limb motor function defined by a Fugl-Meyer Assessment for upper extremity (FMA-UE) score < 55, similar to other studies [40,41]. The exclusion criteria included the following: (1) any additional medical or neurological condition that would affect the ability to comply with the study protocol; (2) any severe visual or language impairment; and (3) pregnancy. The Ferrara Ethics Committee approved the study (protocol code 170294, approved on 13 April 2017), which was registered on ClinicalTrials.gov under the number NCT04622189 and adhered to the CONSORT 2010 statement (Appendix A display CONSORT 2010 Checklist and Flow Diagram) [42]. The objectives, procedures, timelines, risks, and potential benefits of the study, as well as its requirements, were explained to all participants prior to the commencement of the study. All participants signed a written informed consent form. All procedures were performed in accordance with the ethical standards outlined in the Declaration of Helsinki [43].

At baseline (i.e., T0), demographic and clinical information, including stroke severity, as assessed by the National Institute of Health Stroke Scale (NIHSS) [44], was collected, and a neuropsychological assessment was performed. The neuropsychological evaluation included the Cognitive Reserve Index questionnaire (CRIq) and the Test of Attentional Performance (TAP, Version 2.3.1). The first method is used to assess an individual’s cognitive reserve by collecting information from their entire adult life. The questionnaire comprises 20 items divided into three sections, which provide subscores for education, work activity, and leisure time [45]. TAP is a standardized, computer-based assessment of various aspects of attention that is widely used as a diagnostic tool [46]. For the present study, two subtests of the TAP were selected, corresponding to the subtests of divided attention and Go/No-Go. The first assesses the ability to pay attention simultaneously to ongoing processes; in this test, a visual and an auditory task must be processed in parallel. The second evaluates the ability to perform an appropriate reaction under time pressure and simultaneously inhibits an inappropriate behavioral response. The “2 of 5” test form (two critical stimuli among five stimuli) was used; it consists of a sequence of five squares with different patterns appearing on the screen. Two of these squares are defined as target stimuli. Consistent with the literature, impairment on the TAP divided attention and Go/No-Go subtests was defined as performance below the 5th percentile of age- and education-adjusted normative data (collected by the Neuropsychology Service of the Medical Rehabilitation Unit of the Ferrara University Hospital to ensure alignment with the socio-cultural context of the study population), based on reaction times and error rates (omissions and commissions) [47,48,49]. Due to the specific focus of this work, analyses were conducted on reaction times. Moreover, all participants completed Zung’s Anxiety and Zung’s Depression Self-Assessment Scale to assess their mood over the past week, with a focus on anxiety and depressive symptoms [50].

All participants were scheduled to follow an AOT protocol involving four consecutive weeks with five sessions each week. Each session consisted of 3 blocks (sessions) per day, each lasting approximately 15 min, totaling 45 min daily. To measure the attentional level index during the four weeks of training, we used the accuracy of interactive computerized questions (see the next section for further details). Finally, an exploratory engagement questionnaire was created to assess the level of involvement in the treatment. The AOT engagement questionnaire is an 8-item, self-reported scale. It was developed to measure engagement with physical and functional rehabilitation interventions. It includes three categories of items: satisfaction with the treatment, perception of mental fatigue, and level of motivation. Items were generated based on a review of engagement constructs in rehabilitation and refined through discussions with the clinical research team.

The FMA-UE was used as the primary motor outcome measure to evaluate the extent of sensorimotor arm impairment following treatment completion (i.e., T1). This scale is considered the most sensitive for detecting therapeutic changes early after a stroke in patients with arm paresis [51,52,53]. The Box and Block Test (BBT), which assesses manual dexterity [54], and the Barthel Index (BI), an ordinal scale measuring the degree of assistance needed by an individual in mobility and self-care activities for daily living [55,56], were also performed.

### 2.2. AOT Protocol

The participants sat with their arms resting on the table at a distance of 65 cm from a 24-inch LCD screen on which the stimuli were presented. Each AOT session consisted of ten different videos, each repeated four times (40 videos per block), allowing patients to observe 120 videos per day. The videos were randomized over the four weeks of treatment, and the order in which they were presented was kept fixed for all the subjects. The video movements were displayed from a first-person perspective to maximize corticomotor excitability [31]. The movements in the video were performed by a single healthy female actor wearing gloves. The following common categories of actions were included in this study: (a) transitive gestures (i.e., actions performed during spare time, in the office, or while cooking, to more general and straightforward reaching to grasp or pinching); (b) intransitive gestures. The actions were shown with the right or left hand, depending on the patient’s hemiparetic side. Patients were asked to carefully observe the upper limb movement video clips to prepare themselves to imitate the presented actions as accurately as possible. In each block, the following elements were presented in sequence: (1) a fixation period with a countdown (duration: 3000 ms); (2) the start of the video clip with a static image of hands before starting movement (duration: 1000 ms); and (3) the start of movement (duration: variable depending on the video between 4000 and 6000 ms). At the end of video 4, which was repeated as the 16th, 23rd, and 38th stimulus, each subject was asked to imitate the movement.

To maintain concentration during activities and thus maintain an adequate performance level in treatment, interactive computerized questions were created. These include additional questions that request sensorimotor processing, i.e., they are characterized by simultaneously requiring sensory (auditory/visual) and motor processing (such as imagery or active movement). The questions included choice-making, spot-the-differences, order and sequencing, series completion, and identifying the movement and its meaning, which required attention and other cognitive skills (such as working memory and executive functions) [57]. The total number of questions was 120 (2 questions per session). The participants received online feedback for each question. Throughout the entire treatment, the therapist provided verbal instructions and offered assistance when needed. Therefore, these questions were particularly suitable for exploring the interface between MNS (which is essential for sensorimotor integration) and the functions performed by frontoparietal networks, as well as for assessing how AOT improved patients’ performance (i.e., their accuracy in responding to various tasks) during treatment [40,58,59]. Importantly, therapists delivering AOT were blinded to the patients’ attentional classification to prevent potential bias in patient–therapist interactions and to ensure consistent administration of the intervention.

### 2.3. Data Analysis

Baseline clinical and demographic characteristics are expressed as mean and standard deviation (SD) or percentage, depending on the distribution of the variables. Regarding the TAP, all reaction time data are shown as the mean and SD of the raw scores. Based on a clinical evaluation, our sample was categorized into two classes: those with attention deficit (AD) and those without attention deficit (No_AD), according to the scoring criteria established in the literature [46]. Between-group differences were explored using the independent *t*-test. To investigate time effects (i.e., T0 and T1 at the end of the treatment period), paired-sample t tests were applied separately for the AD and No_AD subgroups; the results are reported as mean and 95% confidence interval (CI). For the patient’s engagement questionnaire, the responses were summarized as a mean for each question within each subgroup. The accuracy of attentional questions was calculated separately during the four weeks of training as the mean and SD. A repeated-measures analysis of variance (RM-ANOVA) (*p* < 0.05) with time (4 levels [weeks, W]: W1, W2, W3, W4) as within-subject factors and attention deficit (2 levels: presence, absence) as a between-subject factor was performed to reveal the difference in total accuracy within each group throughout the four weeks of treatment. For this analysis, in cases of significant main effects and/or interactions, Bonferroni-corrected post hoc tests were computed. If the sphericity assumption was violated in Mauchly’s test, the Greenhouse-Geisser correction was applied to the degrees of freedom. Correlations between clinical and demographic variables were tested with the Spearman-Rho test. Statistical significance was set at a level of *p* < 0.05. Statistical analysis was performed with Jamovi Computer Software (Version 1.6) (Jamovi, Sydney, Australia).

## 3. Results

A total of twelve subacute stroke patients were recruited. Two patients were excluded because they did not complete the treatment due to reasons related to the COVID-19 pandemic. Thus, ten patients (median age, 57.9; 70% male; 27.1 days from stroke onset) completed the AOT protocols. Patients were divided into two subgroups: those with attention deficit (AD) (n = 6) and those without attention deficit (No_AD) (n = 4). Both subgroups displayed similar demographic and clinical motor characteristics at baseline, except for hand dominance, where the No_AD group had a majority of left-handed individuals (75% of patients) (see Table 1 for an overview). However, the two subgroups presented statistically significant differences in neurocognitive and psychological variables at baseline. Specifically, the AD group exhibited neglect syndrome and was characterized by lower cognitive reserve indexes. Moreover, they also presented anxious and depressive signs/symptoms and longer reaction times in the Go/No-Go and dual tasks. See Table 2.

The features of training, expressed as the mean values (range), were as follows: 6 sessions per week (3–10), 16.2 days (1–40) of treatment duration, 12.4 min (5–30) per session, and 16.9 min of observed action performance (5–36). Therefore, compared to the planned protocol, the treatment was given at lower doses, which is common in real-world clinical practice [38]. The trend in the accuracy of the questions over the four weeks is shown in Figure 1; both groups showed sustained improvement up to the third week, after which the accuracy decayed.

The difference in total accuracy over the four weeks between the two subgroups reached statistical significance (*p* = 0.036): the No_AD group showed a higher percentage of accuracy (mean 89.6 ± 2.27%) compared with the AD group (mean 82.8 ± 5.00%). When examining accuracy week by week, subgroup differences appeared at week 3 (No_AD: mean 95.0 ± 3.83%; AD: mean 89.2 ± 3.75%; *p* = 0.044) and week 4 (No_AD: mean 88.6 ± 6.74%; AD: mean 77.4 ± 3.26%; *p* = 0.007). The No_AD group exhibited higher accuracy than the AD group between weeks 3 and 4. The RM-ANOVA comparison suggested a main effect of time across the total sample (F(3, 27)= 11.3, *p* ≤ 0.001, ηp^2^ = 0.12), with Bonferroni-corrected post hoc tests indicating accuracy differences between W1–W2 (*p* = 0.036), W1–W3 (*p* = 0.004), and W3–W4 (*p* < 0.001). A second RM-ANOVA indicated a possible group effect (F(1, 8) = 6.34, *p* = 0.036, ηp^2^ = 0.13), and post hoc pairwise comparisons pointed to an influence of attention deficit on accuracy (*p* = 0.036). Post hoc tests for the interaction suggested differences in accuracy between weeks 3 and 4 within each subgroup (AD *p* = 0.004; No_AD *p* = 0.006).

Motor performance for the two subgroups (AD and No_AD) is summarized in Table 3. Both groups showed improvements in motor and functional outcome measures over time, with the No_AD subgroup exhibiting a greater degree of improvement. However, none of the clinical measures reached statistical significance, with the exception of FMA-UE (*p* = 0.013) (see Table 3). The CRI Working Activity score showed a significant correlation with motor improvements addressed by the FMA-UE (rho = 0.786, *p* = 0.048).

All participants in both groups completed the training program. However, the level of engagement significantly differed between subgroups (*p* = 0.013). Specifically, the No_AD group reported, on average, a greater engagement rate (mean 4.38) compared with the AD group (mean 3.13). See Figure 2.

The AD group described AOT as less motivating and stimulating, and perceived a minor sense of accomplishment and gratification. They reported significant difficulties in focusing on the training and defined it as a more challenging, mentally demanding task. Moreover, they described the duration of the daily session as inadequate. Finally, they perceived the training as less fun, less energizing, less pleasurable, and less enjoyable to repeat at home. Figure 3 provides an overview of these findings.

The mean engagement score showed a positive association with the mean accuracy score (rho = 0.9075, *p* ≤ 0.001) and a negative association with attention deficit (rho = −0.873, *p* = 0.002).

## 4. Discussion

Taken together, the results of this pilot study provide preliminary evidence that attentional deficits may affect engagement and functional outcomes in subacute stroke patients undergoing AOT. Although both groups showed similar baseline motor function and completed the intervention with high adherence, patients without attentional deficits (No_AD) tended to perform more accurately during training sessions and reported higher levels of engagement. They also demonstrated greater improvements in upper limb motor function, with FMA-UE gains reaching statistical significance. In contrast, patients with attentional deficits (AD) often reported lower motivation, found the training more cognitively demanding, and described the intervention as less enjoyable. Moreover, engagement levels were positively correlated with training accuracy and attention status, indicating that cognitive factors might influence responsiveness to rehabilitation, as previously demonstrated by our group [60]. However, due to the small sample size and exploratory design, these findings should be interpreted with caution. They are best seen as hypothesis-generating and emphasize the need for larger, well-powered studies to verify whether attentional impairments truly limit both engagement in cognitively demanding rehabilitative protocols and the potential benefits derived from them.

Stroke rehabilitation continues to face the challenge of substantial variability in recovery trajectories, even when patients undergo comparable motor-oriented interventions [61]. Reasonably, some of this variability may arise, at least in part, from overlooking the cognitive domain during neurorehabilitation practice. Consistently, motor and cognitive impairments are often managed as distinct entities, and there is limited evidence regarding their relationship [62,63]. This study contributes to the field by highlighting the impact of attentional performance on both patient engagement and responsiveness to AOT. While attentional deficits are a well-documented consequence of stroke, their role in shaping rehabilitation outcomes has rarely been examined in an explicit and systematic manner [64,65]. Our findings offer new insights into this relationship, suggesting that attentional impairments might restrict patients’ ability to fully engage with AOT and thus limit its therapeutic effectiveness. Indeed, attentional deficits were associated with lower task accuracy during AOT, decreased self-reported engagement, and smaller improvements in upper limb motor recovery. Importantly, although all patients followed the protocol, those with attentional impairments found the treatment less stimulating, more cognitively exhausting, and less rewarding. They reported having more difficulty maintaining focus during daily sessions and felt less accomplished compared to their peers without attentional issues. These results suggest that attentional capacity not only affects objective measures of engagement, such as accuracy on computerized questions, but also significantly influences patients’ subjective experience of rehabilitation. Such perceptions may reduce motivation, effort, and persistence, further exacerbating the effects of attentional difficulties on functional recovery.

The temporal dynamics observed in task accuracy provide further insight into the attentional demands of AOT. Accuracy steadily improved during the first three weeks of training but plateaued and slightly declined in the fourth week. This suggests that engagement and attention reasonably enhanced owing to the proposed treatment, which was engaging and dynamic, and then stabilized and decayed. Nevertheless, the absence of a control group receiving an alternative treatment hinders our ability to attribute the observed differences to the AOT protocol definitively. Moreover, this trend was particularly evident in the subgroup with attentional deficits, suggesting a possible ceiling effect in their ability to maintain focus over extended training durations. Considering that previous reviews have highlighted significant variability in AOT protocols, our findings suggest that session length and program duration should be carefully adjusted to match patients’ attentional resources [29]. Thus, in clinical practice, tailored AOT rehabilitation interventions specifically adapted to cognitive and attentional abilities appear to be relevant in driving subjects’ engagement and motor recovery.

Although our sample was small, 60% of patients with right hemisphere injuries presented attentional deficits, which aligns with literature data [66]. Specifically, the group of patients with attention deficits exhibited poor performance, characterized by longer reaction times, in both the Go/No-Go test and the dual task of the TAP. Furthermore, as in our sample, attention deficits often overlap with other neuropsychological issues, especially neglect [7,67,68,69,70]. These disorders significantly affect daily functioning and hinder successful rehabilitation [9,10]. Thus, a comprehensive assessment is necessary early on to identify the types of attention deficit present after a stroke, allowing for the tailoring of appropriate cognitive rehabilitation. Furthermore, attention deficits significantly impede participation in many daily activities, ultimately affecting other important areas, such as cognitive reserve [71,72,73]. By limiting engagement in mentally stimulating tasks and reducing opportunities for environmental interaction, attentional problems can lead to a gradual depletion of cognitive resources, accelerating functional decline, and reducing overall adaptive capacity. In our study, patients with attention deficits presented a lower cognitive reserve index across all areas: education, work, and leisure. Importantly, the CRIq “working activity” domain showed a significant correlation with FMA-UE increases. Indeed, previous studies underlined the relevant role of cognitive reserve on post-stroke recovery, where higher personal cognitive resources were related to fewer cognitive disorders and overall disability after stroke [74,75]. However, even if our exploratory findings are limited by the inclusion of a comprehensive clinical index and not thoroughly explored or analyzed in consideration of interdependence with other clinical variables, what we observed supports the key role of cognitive resources in determining post-stroke recovery paths, both in terms of attention and motor impairments.

The observed link between attention and motor outcomes aligns with neurophysiological models of AOT. Indeed, action observation activates the MNS, which overlaps with brain regions involved in both motor and cognitive functions, thereby supporting motor execution through attentional control, sensorimotor integration, and performance monitoring [28,76,77,78]. Cognitive resources, particularly attention and memory, are therefore essential to engage effectively with rehabilitation tasks [11]. Evidence shows that early cognitive impairment predicts worse functional outcomes, while intact attentional abilities, particularly visuospatial attention, support motor recovery [23,79]. By leveraging the overlap between motor and cognitive systems, AOT exploits the mechanism where observing an action activates the same neural structures involved in performing it [28]. Crucially, the structured and attention-demanding nature of AOT may boost patients’ ability to stay focused and process sensorimotor feedback across multiple trials. In fact, AOT has been shown to enhance concentration and multitasking skills in stroke patients, with benefits linked to MNS-related regions in the prefrontal cortex and temporal gyrus [80]. However, when attentional deficits are present, this integrative process becomes disrupted, resulting in decreased engagement and worse rehabilitation results. Our findings suggest that attentional resources may play a significant role in shaping responsiveness to AOT, particularly in upper limb recovery as measured by the FMA-UE, which is often regarded as an indicator of true motor recovery during the subacute stroke phase [81]. Although other motor outcomes did not exhibit significant differences between groups, the observed patterns are consistent with the hypothesis that attention can influence rehabilitation trajectories. Taken together, these results tentatively suggest that AOT could benefit both motor and cognitive areas, although larger studies are needed to better understand the extent of these effects.

Furthermore, engagement is increasingly recognized as a key factor in successful rehabilitation, as it enhances neuroplasticity and facilitates functional recovery in patients with neurological disorders [17,82]. In our study, engagement was positively linked to the accuracy of interactive computerized questions, which served as a proxy for concentration—the ability to focus on the task while ignoring distractions [83]. Consistently, engagement has been described as a multidimensional construct that includes both observable behaviors and interpersonal processes, and is affected by patients’ emotional state and cognitive functioning [13,16,84]. As previously mentioned, analysis of accuracy over the four weeks of training showed an initial increase in attention and engagement, especially between W1 and W3, followed by a drop in W4. This pattern may reflect the longer-than-average duration of our protocol, which could have caused monotony and mental fatigue, leading to decreased commitment over time. These findings are consistent with previous evidence suggesting that videos around 5–6 min long may provide the optimal balance between maintaining attention and maximizing training effectiveness [29]. Notably, subgroup analyses revealed that patients without attention deficits consistently performed better than those with deficits, with significant differences emerging in W3 and W4, further supporting the connection between attentional capacity and sustained engagement. Nevertheless, overall adherence to AOT was high, with no participants dropping out, underscoring the feasibility and acceptability of the intervention. Importantly, patients with attention deficits not only performed less accurately but also reported lower engagement, describing the training as less motivating, more mentally exhausting, and less enjoyable. Such perceptions likely reflect the impact of attentional limitations on motivation and self-efficacy, reducing the sense of accomplishment and gratification derived from training. This observation aligns with evidence that higher levels of attention and interest in rehabilitation promote greater patient empowerment, knowledge, and adherence, ultimately leading to functional improvements [19,85,86,87,88]. Taken together, these findings suggest that engagement may not be just a secondary factor, but a key mediator connecting attentional capacity to rehabilitation outcomes.

Finally, in our cohort, patients with attentional deficits also exhibited higher levels of anxiety and depression, consistent with evidence that stroke survivors often experience psychological distress due to disability, reduced motivation, and low self-esteem [89,90]. Such symptoms are known to impact executive function, memory, processing speed, and motor performance, and have been linked to lower education as a risk factor for post-stroke depression, despite differences across studies [91,92]. Indeed, psychiatric difficulties may further impede recovery by leading to unhealthy lifestyle choices or decreased adherence to rehabilitation [93]. These findings reinforce the view that psychological well-being and cognitive support are not only vital for quality of life but also play a crucial role in rehabilitation outcomes [94,95,96].

Although we mainly discussed studies supporting our interpretation, it’s important to recognize that other explanations could still be plausible. For example, activation of the MNS during AOT may itself, through the sensorimotor system’s role in simulating observed actions, explain some of the observed gains independently of attentional factors [97]. Additionally, elements intrinsic to the selected AOT protocol—such as the meaningfulness and transitivity of the observed actions—may have affected the outcomes of interest, although the limited standardization of AOT protocols in the literature makes direct comparisons difficult [29,98]. Furthermore, recent reviews have highlighted that long-term rehabilitation outcomes are not solely dependent on attention, but may also rely on memory or other executive functions, factors that could have shaped our results but were not specifically examined in this study [99]. Taken together, these considerations warrant caution when interpreting our findings, as other explanations are still plausible and haven’t been thoroughly investigated in this context to be confidently dismissed.

Our experiment has several limitations that can be addressed through further research. A significant limitation is the premise that the term’ attention’ is not a unitary construct, as it encompasses a subset of processes and mechanisms that require further study, especially to provide a single, unified conceptualization [100]. However, based on our findings and supporting evidence from the literature, we believe that we have examined the attention domains most relevant to the outcomes of interest, especially within the context of the neurorehabilitation approach under study in a highly applicable manner to clinical practice [25,101,102]. Nevertheless, future scientific investigations employing multidimensional neuropsychological tools capable of broader analysis and quantitative detection of attention subsets may expand upon and further delineate our exploratory findings. Another limitation of this pilot study is the imbalance in handedness between groups, as the No_AD group included a predominance of left-handed individuals. This may have introduced a potential confounding effect, especially in a visuomotor training context involving unilateral video stimuli [103,104]. Furthermore, in our sample, neglect, mood symptoms, and right-hemisphere lesions were more prevalent in the AD group. These factors are known to influence both cognitive and motor outcomes after stroke and may have contributed to the observed group differences. Specifically, spatial neglect adversely affects rehabilitation outcomes, leading to slower functional gains, higher risk of falls, and longer hospital stays [105]. Additionally, common post-stroke mood disorders like depression, apathy, and anxiety negatively affect both cognitive and functional recovery, contributing to increased disability and poorer outcomes [106]. Given the limited sample size, we were unable to apply statistical models to control for potential confounders or to fully explore the complex, multifactorial relationships between attentional deficits, cognitive reserve, and other relevant outcomes [107]. Future research should address this limitation, acknowledging the important role of this metric in both physiological and pathological conditions. Nonetheless, it is important to emphasize that this study was designed as a pilot investigation to optimize a rehabilitative strategy—specifically, AOT—during a period when access to traditional rehabilitation was severely limited due to the COVID-19 pandemic. Therefore, the limited sample size was not a methodological oversight but rather a direct result of the public health emergency that occurred during the study period. While larger studies will undoubtedly be needed to confirm the generalizability of these findings and to offer a more detailed understanding of the outcomes of interest, we believe that, on the basis of previous evidence from our group and others using MNS-based paradigms, the recruited cohort provides a valid and meaningful foundation for an initial investigation of the proposed hypotheses [27,108,109]. Aside from the attentional domain, the patients underwent customized neuropsychological assessments that limited the standardized evaluation of a broader range of cognitive areas. Furthermore, a neuropsychological evaluation at the end of treatment to identify any potential improvements in both the functional and cognitive areas was lacking. Moreover, it would be helpful to assess the impact of attention deficits and patients’ engagement level by comparing outcomes with a control group undergoing a conventional or different type of intervention to evaluate AOT more effectively. Another limitation concerns the AOT engagement questionnaire, which was specifically developed for this study. Although designed to capture satisfaction, fatigue, and motivation, the instrument has not undergone formal psychometric validation, which restricts the interpretability of the engagement results. Future research should address this gap by evaluating the reliability and validity of the scale in larger and more diverse cohorts. In addition, future research should gather data on a wider variety of neurophysiological attention markers, as employing neurofeedback techniques would increase the applicability of our results. Indeed, a real-time electroencephalography biomarker for attention and engagement could provide information about the temporary functional changes caused by motor treatment sessions, and integrating behavioral and neurophysiological data is a valuable approach for understanding and customizing upper limb motor therapy in stroke patients, as well as for developing promising neuromodulation treatments [110,111,112,113]. Finally, since our sample included patients in the subacute stroke phase, it is likely that some of the observed improvements are due to the natural recovery process that usually occurs during this period—mainly driven by increased neuroplasticity in the early post-acute stage—instead of being solely caused by the proposed intervention [114]. However, without a control group, it remains difficult to disentangle the specific effects of the proposed intervention from those attributable to the intrinsic recovery process. Therefore, future studies should explore whether these findings apply to other stages of stroke, such as the chronic phase, and include a conventional treatment group to better clarify their implications for long-term neurorehabilitation strategies.

## 5. Conclusions

This pilot study examined the impact of attentional performance on engagement and motor recovery during an AOT rehabilitation program for subacute stroke patients. Despite the limited sample size, our findings offer preliminary evidence that patients without attentional deficits generally engaged more effectively with the training, demonstrated higher accuracy in interactive tasks, and attained greater improvements in upper limb function—most notably reflected in the FMA-UE, the sole outcome measure exhibiting significant group differences and serving as a pertinent tool for identifying motor restitution following stroke [114]. Conversely, patients with attentional deficits often reported decreased motivation and increased cognitive effort, which may have limited their functional improvements. These results should be interpreted with caution due to the exploratory design and small sample size; however, they underscore the potential importance of attentional resources in influencing responsiveness to cognitively demanding rehabilitative protocols, such as AOT. For clinical practice, this indicates that evaluating and supporting attention may enhance both engagement and treatment outcomes, which are needed to coherently provide appropriate and tailored rehabilitation interventions. On the other hand, it also suggests that AOT could potentially be used, although the absence of a control group prevents us from confirming this at present, not only to improve motor outcomes in patients with particularly severe motor impairments (i.e., unable to undergo conventional rehabilitation), as already suggested, but also to enhance their attentional performance, which is often impaired in this population [58]. This emphasizes the two-way link between AOT and attentional skills. Larger, well-controlled studies are necessary to verify these findings and determine if adding cognitive screening and personalized support can improve AOT’s effectiveness in stroke rehabilitation.

## Figures and Tables

**Figure 1 jcm-14-06618-f001:**
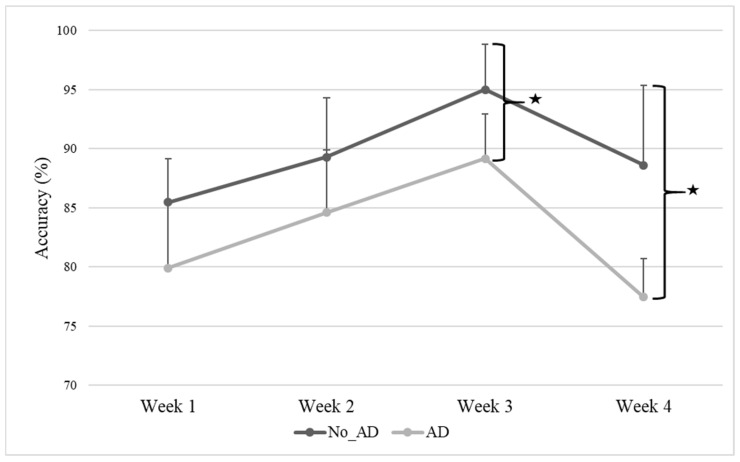
Accuracy of interactive computerized exercises during the four weeks of AOT. Abbreviation: No_Attention Deficit (No_AD); Attention Deficit (AD). Description: Data are reported as mean and standard deviation for each week. Changes in accuracy of the two groups across the four weeks of AOT. Accuracy for each week of treatment was significantly different between the two subgroups in week 3 (No_AD: mean 95.0 ± 3.83%; AD: mean 89.2 ± 3.75%; *p* = 0.044) and week 4 (No_AD: mean 88.6 ± 6.74%; AD: mean 77.4 ± 3.26%; *p* = 0.007) as indicated by the stars.

**Figure 2 jcm-14-06618-f002:**
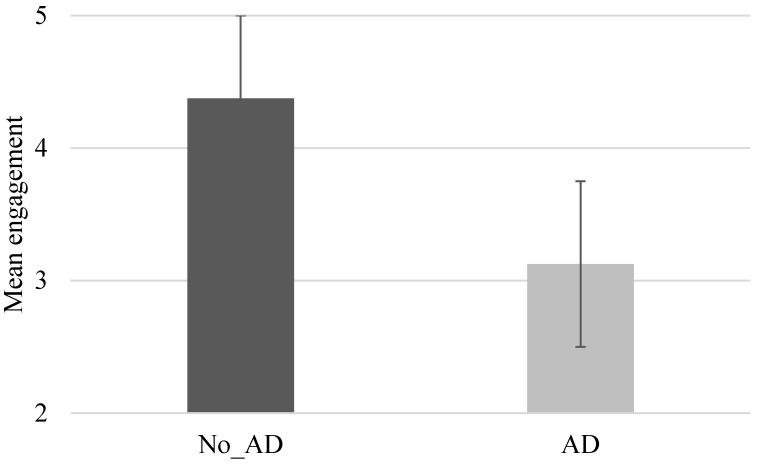
Level of engagement. Abbreviation: No_Attention Deficit (No_AD); Attention Deficit (AD). Description: Level of engagement in the two groups (No_AD: mean 4.38 ± 0.42%; AD: mean 3.2 ± 0.35%; *p* = 0.013).

**Figure 3 jcm-14-06618-f003:**
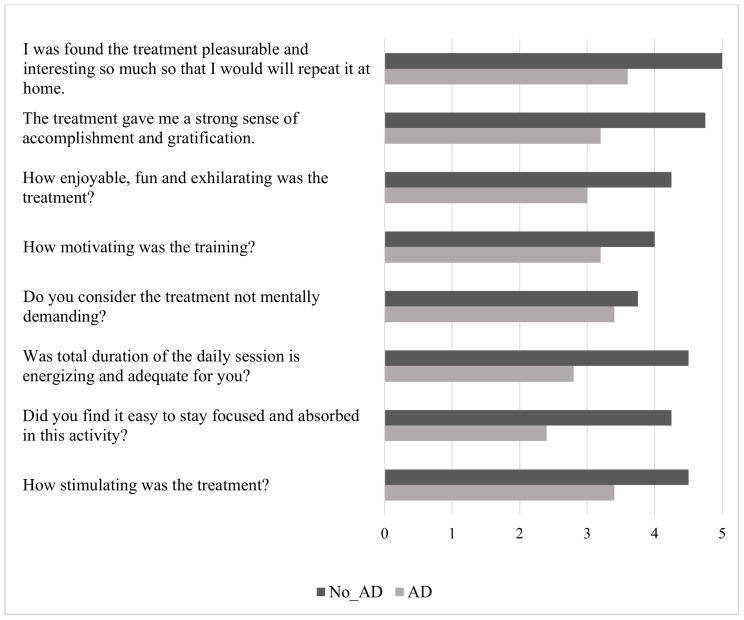
AOT engagement questionnaire. Abbreviation: No_Attention Deficit (No_AD); Attention Deficit (AD). Description: Subjective reports of engagement as rated by the participants on a Likert Scale (0 not at all–5 very much). Means of each item of the AOT engagement questionnaire in the two groups (No_AD and AD) are presented.

**Table 1 jcm-14-06618-t001:** Demographic and clinical physical data of the sample at baseline.

Measures	Total Sample (N = 10)	No_AD Group (N = 4)	AD Group (N = 6)	*p* Value
Stroke severity (mild,%)	5 (50%)	3 (75%)	2 (33.33%)	0.242
Sex (male,%)	7 (70%)	3 (75%)	4 (66.66%)	0.807
Age (mean, SD)	57.9 (9.22)	56.5 (4.20)	58.8 (11.8)	0.719
Stroke type (hemorrhagic, %)	6 (60%)	3 (75%)	3 (50%)	0.486
Dominant hand (left, %)	3 (30%)	3 (75%)	0 (0%)	0.005 *
NIHSS entry (mean, SD)	12.6 (3.24)	10.3 (2.89)	13.7 (3.01)	0.157

Abbreviations: No_Attention Deficit (No_AD); Attention Deficit (AD); NIHSS (National Institutes of Health Stroke Scale); SD (standard deviation); N (number of subjects). * *p* < 0.05.

**Table 2 jcm-14-06618-t002:** Neuropsychological and psychological clinical data of the sample at baseline.

Measures	Total Sample (N = 10)	No_AD Group (N = 4)	AD Group (N = 6)	*p* Value
TAP: Go/Nogo_Reaction Time (mean, SD) [95%CI]	591 (130)	444.07 (40.57) [343.29, 544.85]	678.95 (52.37) [613.93, 743.97]	<0.001 *
TAP: Dual task_Reaction Time (mean, SD) [95%CI]	815 (185)	667.64 (20.74) [638.34, 696.95]	925.63 (173.98) [712.78, 1138.49]	0.051
Anxious and depressive signs/symptoms (n, %)	4 (40%)	0 (0%)	4 (66.66%)	0.035 *
Neglect (n, %)	4 (40%)	0 (0%)	4 (66.66%)	0.035 *
CRIq (mean, SD) [95%CI]	105 (17.5)	120 (14.9) [102.85, 137.15]	92.2 (4.09) [87.99, 96.41]	0.005 *
CRI-Education (mean, SD) [95%CI]	96.2 (14.9)	107.5 (8.13) [88.75, 126.25]	87.2 (1.71) [83.25, 91.15]	0.029 *
CRI-Working Activity (mean, SD) [95%CI]	102 (12.1)	110.75 (6.57) [95.59, 125.91]	94.8 (1.98) [90.22, 99.38]	0.037 *
CRI-Leisure Time (mean, SD) [95%CI]	112 (16.7)	126.25 (12.8) [111.44, 141.06]	100.4 (7.83) [92.33, 108.47]	0.007 *

Abbreviations: No_Attention Deficit (No_AD); Attention Deficit (AD); Cognitive Reserve Index questionnaire (CRIq); SD (standard deviation); N (number of subjects); TAP: Test of Attentional Performance. * *p* < 0.05.

**Table 3 jcm-14-06618-t003:** Mean (±SD) and significance of rehabilitative gains after AOT.

Measures	Total Sample (N = 10)	No_AD Group (N = 4)	AD Group (N = 6)	*p* Value
∆ FMA-UPPER LIMB (mean, SD) [95%CI]	6.52 (4.680)	10.8 (5.68) [4.33, 17.17]	3.33 (1.21) [2.21, 4.45]	0.013 *
∆ FMA-WRIST (mean, SD) [95%CI]	2.75 (3.178)	4.25 (3.10) [0.75, 7.75]	1.33 (3.33) [−1.74, 4.41]	0.201
∆ FMA-HAND (mean, SD) [95%CI]	3.81 (4.398)	5.75 (6.29) [−1.37, 12.87]	1.83 (3.06) [−0.99, 4.66]	0.219
∆ FMA total (mean, SD) [95%CI]	13.53 (11.772)	21.0 (16.4) [2.41, 39.59]	7.33 (6.25) [1.56, 13.11]	0.096
∆ BBT—PARETIC LIMB (mean, SD) [95%CI]	11.34 (12.500)	13.8 (12.0) [0.17, 27.33]	8.00 (15.0) [−5.06, 22.06]	0.540
∆ BI (mean, SD) [95%CI]	25.49 (20.99)	38.8 (26.9) [8.34, 69.16]	15.8 (16.6) [0.54, 31.12]	0.130

Abbreviations: No_Attention Deficit (No_AD); Attention Deficit (AD); Barthel Index (BI); Box and Block Test (BBT); Fugl-Meyer Assessment (FMA); SD (standard deviation). * *p* < 0.05.

## Data Availability

The data that support the findings of this study are available from the first author (G.M.) upon reasonable request.

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
