# Peer review of "Action Observation Training for Upper Limb Stroke Rehabilitation: A Pilot Study on the Role of Attention"

_jcm, 2025, doi:10.3390/jcm14186618_

Round 1
Reviewer 1 Report
Comments and Suggestions for Authors
Title
The manuscript's title emphasizes an important aspect, but that wasn't controlled in the research: engagement. I suggest removing the word 'engagement' from the title, leaving the subtitle as simply: A Pilot Study on the Role of Attention.
Introduction
A rationale is needed to explain the relationship between AOT and Attentional Deficits.
Methods:
What was the criterion to include patients in both groups (with or without attentional deficits)?
Data analysis is particularly complex for a pilot study involving a sample of 4-6 per group.
What is the rationale for running regressions for the accuracy and engagement stages? Why is this analysis important? What outcomes were used as independent variables? Was cognitive reserve included in these outcomes?
Furthermore, regression analysis is not suitable for very small samples. I think you should be excluding this analysis, its results, discussion, and conclusion.
The trial register (NCT04622189) previously reported more outcomes than those reported in the manuscript. Outcomes not reported were: Modified Ashworth Scale, Visual Analogue Scale, EEG power in the alpha band, Motion Capture of trunk and affected hand movements, and Oxford Cognitive Screen. Why were these outcomes not reported?
How was accuracy calculated? By percentage of errors? Why is this outcome important for the patients and the theme studied?
Results
Lines 215-217 contain information about the characteristics of the therapy, which differ from what is proposed without justification.
Discussion
The relationship between attention deficit and other variables is discussed more than the influence of this impairment on therapeutic outcomes, engagement, and accuracy in AOT. I suggest minimizing the parts of the discussion that relate attention deficit to other outcomes and emphasizing its relationship and influence on AOT.
The main limitation of the study was its inability to control the patients' overall cognitive reserve. In this sense, the difference between the groups may have been due not to attention deficit per se, but to decreased cognitive reserve, which creates a fragility in the observed relationships.
Author Response
Title
The manuscript's title emphasizes an important aspect, but that wasn't controlled in the research: engagement. I suggest removing the word 'engagement' from the title, leaving the subtitle as simply: A Pilot Study on the Role of Attention.
First, we thank the Reviewer for taking the time to review our manuscript and for providing valuable suggestions to improve our work. Regarding the title, although our primary focus was on attention, we also examined engagement with a specific questionnaire, as detailed in the Objectives and Methods sections. However, we agree that it is important to emphasize the main point of the work, which is attention. Therefore, we have revised the title as the Reviewer suggested.
New title: “Action Observation Training for Upper Limb Stroke Rehabilitation. A pilot study on the role of attention”
Introduction
A rationale is needed to explain the relationship between AOT and Attentional Deficits.
We would like to express our gratitude to the Reviewer for this valuable observation, which has contributed to enhancing the quality of our work and providing additional reasoning for our key points. Accordingly, we expanded the Introduction section as follows (please see the Introduction section, lines 90-101):
“Importantly, since AOT depends on activating the MNS and related frontoparietal networks, it is closely connected with attentional control systems [32]. Therefore, successful engagement in AOT requires focusing attention to observe, process, and mentally simulate the watched motor actions [33]. Moreover, directing visual attention explicitly during AOT facilitates corticospinal excitability and may accelerate motor relearning through observation [34]. Attention deficits may limit the ability to properly encode and integrate observed movements, thereby reducing the effectiveness of AOT [35]. On the other hand, structured AOT protocols might also help train attention skills, as patients must repeatedly stay focused and selectively process relevant motor cues [36]. Thus, attention is both a necessary prerequisite and a potential focus in AOT, and understanding this relationship is essential for optimizing patient selection and therapeutic results”
References:
- Nummenmaa L, Smirnov D, Lahnakoski JM, Glerean E, Jääskeläinen IP, Sams M, Hari R. Mental action simulation synchronizes action-observation circuits across individuals. J Neurosci. 2014 Jan 15;34(3):748-57. doi: 10.1523/JNEUROSCI.0352-13.2014. PMID: 24431433; PMCID: PMC3891955.
- Kemmerer D. What modulates the Mirror Neuron System during action observation?: Multiple factors involving the action, the actor, the observer, the relationship between actor and observer, and the context. Prog Neurobiol. 2021 Oct;205:102128. doi: 10.1016/j.pneurobio.2021.102128. Epub 2021 Jul 31. PMID: 34343630.
- Sánchez Silverio V, Abuín Porras V, Rodríguez Costa I, Cleland JA, Villafañe JH. Effects of action observation training on the walking ability of patients post stroke: a systematic review. Disabil Rehabil. 2022 Dec;44(24):7339-7348. doi: 10.1080/09638288.2021.1989502. Epub 2021 Oct 13. PMID: 34644226.
- D’Innocenzo, G., Gonzalez, C.C., Nowicky, A.V., Williams, A.M., Bishop, D.T., 2017. Motor resonance during action observation is gaze-contingent: A TMS study. Neuropsychologia 103, 77–86. https://doi.org/10.1016/j.neuropsychologia.2017.07.017
- Caligiore D, Mustile M, Fineschi A, Romano L, Piras F, Assogna F, Pontieri FE, Spalletta G, Baldassarre G. Action Observation With Dual Task for Improving Cognitive Abilities in Parkinson's Disease: A Pilot Study. Front Syst Neurosci. 2019 Feb 11;13:7. doi: 10.3389/fnsys.2019.00007. PMID: 30804762; PMCID: PMC6378302.
Methods:
What was the criterion to include patients in both groups (with or without attentional deficits)?
We thank the Reviewer for highlighting this important point, and we entirely agree on the importance of clarifying it. As outlined in the Methods section, patients were categorized into groups with or without attentional deficits based on their performance on two subtests of the Test of Attentional Performance (TAP), specifically divided attention and Go/No-Go. Specifically, patients were classified as “without deficits” only if their performance in both subtests fell within the normal range, while an impairment in even one of the two subtests led to classification in the “with attentional deficits” group. This approach was chosen to increase the accuracy of classification for patients without deficits, as previously suggested by other studies. To clarify this important point, we added the following to our manuscript (please see subsection 2.1, lines 156-162):
“Consistent with the literature, impairment on the TAP divided attention and Go/No-Go subtests was defined as performance below the 5th percentile of age- and education-adjusted normative data (collected by the Neuropsychology Service of the Medical Rehabilitation Unit of the Ferrara University Hospital to ensure alignment with the socio-cultural context of the study population), based on reaction times and error rates (omissions and commissions) [47–49]. Due to the specific focus of this work, analyses were conducted on reaction times.”
We hope this clarification addresses the Reviewer’s concern and, at the same time, improves the overall quality and transparency of our work. We are sincerely grateful to the Reviewer for pointing out this important aspect.
References:
- Schmidt SL, Boechat YEM, Schmidt GJ, Nicaretta D, van Duinkerken E, Schmidt JJ. Clinical Utility of a Reaction-Time Attention Task in the Evaluation of Cognitive Impairment in Elderly with High Educational Disparity. J Alzheimers Dis. 2021;81(2):691-697. doi: 10.3233/JAD-210151. PMID: 33814451.
- Schmidt GJ, Boechat YEM, van Duinkerken E, Schmidt JJ, Moreira TB, Nicaretta DH, Schmidt SL. Detection of Cognitive Dysfunction in Elderly with a Low Educational Level Using a Reaction-Time Attention Task. J Alzheimers Dis. 2020;78(3):1197-1205. doi: 10.3233/JAD-200881. PMID: 33136095.
- Schumacher R, Halai AD, Lambon Ralph MA. Attention to attention in aphasia - elucidating impairment patterns, modality differences and neural correlates. Neuropsychologia. 2022 Dec 15;177:108413. doi: 10.1016/j.neuropsychologia.2022.108413. Epub 2022 Nov 3. PMID: 36336090; PMCID: PMC7614452.
Data analysis is particularly complex for a pilot study involving a sample of 4-6 per group.
What is the rationale for running regressions for the accuracy and engagement stages? Why is this analysis important? What outcomes were used as independent variables? Was cognitive reserve included in these outcomes?
Furthermore, regression analysis is not suitable for very small samples. I think you should be excluding this analysis, its results, discussion, and conclusion.
We thank the Reviewer for highlighting this significant point, and we agree. In fact, we endeavored to utilize these analyses to investigate the complex relationship between the variables of interest in the most appropriate manner, with the aim of proposing potential evidence for future research. Nonetheless, we agree that, considering the sample size and the exploratory nature of the study, such analyses are inappropriate and the results may be misleading. Therefore, also following the suggestion from other Reviewers, we have removed references to regressions throughout the entire manuscript (please see the revised version). We appreciate the Reviewer for helping us improve the scientific accuracy of our work.
The trial register (NCT04622189) previously reported more outcomes than those reported in the manuscript. Outcomes not reported were: Modified Ashworth Scale, Visual Analogue Scale, EEG power in the alpha band, Motion Capture of trunk and affected hand movements, and Oxford Cognitive Screen. Why were these outcomes not reported?
We appreciate the Reviewer’s thoughtful comment and welcome the chance to clarify. Please note that the NCT number mentioned corresponds to a large, long-term study that examined various aspects of post-stroke motor recovery from clinical, neuropsychological, and neurophysiological perspectives. Therefore, not all patients completed every assessment. Indeed, some participants declined electroencephalography recordings or neuropsychological tests but agreed to other evaluations. However, the data we considered essential for this study were available for all included patients. It is also worth noting that one study based on this trial has already been published, and others are currently under review, each focusing on a specific area. Please see, for instance:
- Boni S, Galluccio M, Baroni A, et al. Action Observation Therapy for Arm Recovery after Stroke: A Preliminary Investigation on a Novel Protocol with EEG Monitoring. J Clin Med. 2023;12(4):1327. Published 2023 Feb 7. doi:10.3390/jcm12041327
In line with this approach, the present work specifically examines the role of attention in the subset of patients for whom relevant data were available, thereby avoiding unnecessary duplication of previously reported findings while providing, hopefully, a new and clinically meaningful contribution to the understanding of recovery processes.
How was accuracy calculated? By percentage of errors? Why is this outcome important for the patients and the theme studied?
We sincerely thank the Reviewer for this important observation, which provides us with an opportunity to better explain the rationale behind our methodology choice. Specifically, as detailed in the Methods section, the interactive questions were designed to encompass a range of tasks that integrate sensorimotor processing with higher-order executive functions, such as attention and cognitive flexibility, with performance assessed in terms of accuracy (i.e., the percentage of correctly completed tasks). There were several reasons for including these questions. Initially, by integrating sensorimotor functions with executive processes, we regarded them as particularly appropriate for evaluating the interface between the mirror neuron system and attentional mechanisms—an aspect central to the present investigation. Secondly, by monitoring patients’ accuracy throughout the treatment weeks, we were able to investigate whether AOT could facilitate enhancements not only in motor performance but also in attentional engagement, potentially indicating stimulation of both sensorimotor circuits and frontal-parietal attentional networks. Finally, this approach provided an additional measure of patients’ active engagement, as the tasks were varied and required continuous concentration and focus during AOT administration. Taken together, we believe these considerations justify using interactive question accuracy as a meaningful, easy-to-collect, and scalable measure within our study design. However, we agree with the Reviewer on the relevance of clarifying this essential point. Accordingly, we added the following to our manuscript (please see the Methods section, lines 215-219):
“Therefore, these questions were particularly suitable for exploring the interface between MNS (which is essential for sensorimotor integration) and the functions performed by frontoparietal networks, as well as for assessing how AOT improved patients' performance (i.e., their accuracy in responding to various tasks) during treatment [40,58,59].”
References:
- Errante A, Saviola D, Cantoni M, Iannuzzelli K, Ziccarelli S, Togni F, Simonini M, Malchiodi C, Bertoni D, Inzaghi MG, Bozzetti F, Menozzi R, Quarenghi A, Quarenghi P, Bosone D, Fogassi L, Salvi GP, De Tanti A. Effectiveness of action observation therapy based on virtual reality technology in the motor rehabilitation of paretic stroke patients: a randomized clinical trial. BMC Neurol. 2022 Mar 22;22(1):109. doi: 10.1186/s12883-022-02640-2. PMID: 35317736; PMCID: PMC8939064.
- Buccino G. Action observation treatment: a novel tool in neurorehabilitation. Philos Trans R Soc Lond B Biol Sci. 2014 Apr 28;369(1644):20130185. doi: 10.1098/rstb.2013.0185. PMID: 24778380; PMCID: PMC4006186.
- Boni, S., Galluccio, M., Baroni, A., Martinuzzi, C., Milani, G., Emanuele, M., Straudi, S., Fadiga, L., Pozzo, T., 2023. Action Observation Therapy for Arm Recovery after Stroke: A Preliminary Investigation on a Novel Protocol with EEG Monitoring. Journal of Clinical Medicine 12, 1327. https://doi.org/10.3390/jcm12041327
Results
Lines 215-217 contain information about the characteristics of the therapy, which differ from what is proposed without justification.
We appreciate the Reviewer for highlighting this discrepancy. Indeed, in the Methods section, we described the planned training protocol, which included four consecutive weeks of AOT, five days a week, with three blocks of about 15 minutes each per day (roughly 45 minutes daily). Nevertheless, as is frequently observed in clinical practice, the implementation of training varied among patients owing to individual clinical conditions, tolerance levels, or organizational challenges such as intercurrent medical complications or scheduling difficulties. Therefore, in the Results section, we reported the actual features of the training to provide an accurate view of how the intervention was implemented in the study population. We believe that showing both the planned protocol and the actual training delivered enhances the transparency of our report and highlights the practical challenges of rehabilitation trials. However, we agree on the importance of clarifying this relevant point. Therefore, we revised our Methods as follows (please see the Methods section, lines 166-167):
“All participants were scheduled to follow an AOT protocol involving four consecutive weeks with five sessions each week.”
Moreover, in the Results section, we added the following clarification to highlight this relevant point (please see the Results section, lines 268-270):
“Therefore, compared to the planned protocol, the treatment was given at lower doses, which is common in real-world clinical practice [38]”.
Reference:
- Borges LR, Fernandes AB, Oliveira Dos Passos J, Rego IAO, Campos TF. Action observation for upper limb rehabilitation after stroke. Cochrane Database Syst Rev. 2022 Aug 5;8(8):CD011887. doi: 10.1002/14651858.CD011887.pub3. PMID: 35930301; PMCID: PMC9354942.
Once again, we thank the Reviewer for emphasizing this important point and helping us improve our manuscript.
Discussion
The relationship between attention deficit and other variables is discussed more than the influence of this impairment on therapeutic outcomes, engagement, and accuracy in AOT. I suggest minimizing the parts of the discussion that relate attention deficit to other outcomes and emphasizing its relationship and influence on AOT.
We sincerely thank the Reviewer for this insightful comment. In accordance with this and feedback from the other Reviewers, we have thoroughly revised the Discussion to better highlight the main findings of our study (please see the revised version of the Discussion). Specifically, we have minimized digressions about the broader links between attention deficits and secondary outcomes and focused more clearly on their connection to therapeutic results, engagement, and accuracy within the AOT protocol. Additionally, we aimed to offer a more balanced interpretation of the results by considering alternative explanations, avoiding unnecessary overlap with the Introduction, and streamlining the narrative to emphasize the most novel and relevant contributions of our work. We hope these revisions have enhanced the clarity and focus of the Discussion as suggested.
The main limitation of the study was its inability to control the patients' overall cognitive reserve. In this sense, the difference between the groups may have been due not to attention deficit per se, but to decreased cognitive reserve, which creates a fragility in the observed relationships.
We thank the Reviewer for this insightful comment, and we fully agree. Indeed, cognitive reserve likely influenced the outcomes of interest. Our study's small sample size prevents us from applying statistical models that can adequately account for this important factor. Indeed, although it is well established that a higher cognitive reserve is predictive of improved cognitive-motor outcomes following an injury, it has also been demonstrated that this factor does not operate in isolation but rather in conjunction with other variables, including age and the nature of the injury. Therefore, considering these interactions, it would have been essential to have much larger sample sizes. Currently, we can only observe that, as expected, the group with attention deficits indeed had a lower cognitive reserve. We fully recognize, however, the importance of this issue, and future studies with larger samples will need to address it using appropriate statistical models. Therefore, we revised our manuscript to highlight these relevant points. Specifically, please see the Discussion section (lines 509-513:
“Given the limited sample size, we were unable to apply statistical models to control for potential confounders or to fully explore the complex, multifactorial relationships between attentional deficits, cognitive reserve, and other relevant outcomes [108]. Future research should address this limitation, acknowledging the important role of this metric in both physiological and pathological conditions.”
We are grateful to the Reviewer for emphasizing this crucial point, which we believe further strengthens the interpretation and future perspectives of our work.
Reference:
Contador I, Alzola P, Stern Y, de la Torre-Luque A, Bermejo-Pareja F, Fernández-Calvo B. Is cognitive reserve associated with the prevention of cognitive decline after stroke? A Systematic review and meta-analysis. Ageing Res Rev. 2023 Feb;84:101814. doi: 10.1016/j.arr.2022.101814. Epub 2022 Dec 5. PMID: 36473672.
Reviewer 2 Report
Comments and Suggestions for Authors
This study focuses on the application of action observation training in upper limb rehabilitation for stroke patients, exploring the impact of attention deficits and patient engagement on rehabilitation outcomes. The research topic addresses the clinically significant issue of functional recovery and cognitive impairments following stroke, demonstrating both innovation and practical relevance. However, the manuscript also contains several major issues.
Abstract
1. Clearly define group assignments.
2. Include mean ± SD or effect size.
3. Weaken causal inferences and avoid absolute conclusions.
Introduction
1. The introduction lists a large number of prevalence rates, manifestations, and risks associated with attention deficits (approximately 46–92% of patients, fall risk, mobility/ADL impact, etc.), but lacks summary and critical thinking.
2. The three-step logic of “what existing studies have done—what is missing—what this study addresses” is absent.
3. The final sentence of the introduction merely states, “We conducted a pilot study... to examine the role of attention and engagement in motor recovery following AOT,” which is overly broad.
4. The introduction repeatedly describes the hazards of attention deficits and the importance of engagement, which highly overlaps with the abstract.
Methods and Results
1. In such a small sample size, any significant results may be due to chance, especially when reporting “significant predictors” in multiple regression analysis, which has extremely low credibility.
2. The absence of confidence intervals (95% CI) and effect sizes fails to meet the standards of high-quality clinical research.
3. Overemphasis on statistical significance.
4. In the results section, the authors include explanatory language (e.g., “training became monotonous, leading to decreased accuracy” and “patients with attention deficits felt more fatigued”), which should belong in the discussion rather than the results.
5. The legends for Figures 1, 2, and 3 are too general and lack key numerical values.
Discussion
1. The authors repeatedly emphasize that “attention deficits significantly hinder rehabilitation,” but in fact, only the FMA-UE metric showed significant differences, while other primary metrics did not.
2. In a small sample (n=10) + pilot study context, such statements imply overly strong causality.
3. It is recommended to soften the tone, e.g., “The results suggest that there may be... which requires validation with a larger sample.”
4. While numerous supporting studies are cited, there is insufficient discussion of “inconsistent results or counterexamples.”
5. This leads readers to perceive the discussion as overly one-sided, lacking academic balance.
6. The discussion section repeats the harm of attention deficits and the importance of engagement (already covered in the introduction), resulting in redundant information.
7. The discussion should focus more on “the novel findings and value of this study” and avoid lengthy background reviews.
Conclusion
1. However, this study is only a pilot study with n=10, and only the FMA-UE result was significant, while most other functional indicators were not significant. The tone of the conclusion does not align with the actual evidence from the study, appearing overly strong.
2. Although the methods section states that the study is a pilot study, the conclusion does not reiterate that “the results are preliminary explorations and should be interpreted with caution.” This may lead readers to mistakenly believe that the results can be directly generalized.
3. High-quality journals typically recommend briefly noting at the end of the conclusion that “further validation is needed in large-scale controlled studies.”
4. The conclusion does not further elaborate on “what this means for rehabilitation practice.”
Comments on the Quality of English Language
Author Response
This study focuses on the application of action observation training in upper limb rehabilitation for stroke patients, exploring the impact of attention deficits and patient engagement on rehabilitation outcomes. The research topic addresses the clinically significant issue of functional recovery and cognitive impairments following stroke, demonstrating both innovation and practical relevance. However, the manuscript also contains several major issues.
We would like to thank the Reviewer for taking the time to evaluate our work and for providing helpful feedback. Specifically, following the important suggestions provided by the Reviewer, we have revised our manuscript, aiming to address all the points raised. We hope this has enhanced the quality of our work, and we thank the Reviewer once again for their valuable opportunity.
Abstract
- Clearly define group assignments.
We thank the Reviewer for highlighting this relevant point, and we agree on the relevance of providing this crucial information. Therefore, we revised our Abstract as follows (please see the revised version of the Abstract, lines 22-25):
“At baseline, they were divided into two subgroups based on attentional performance, as determined by scores on the Test of Attentional Performance (subtests of divided attention and Go/No-Go): those with attention deficits (AD, i.e., deficits in one or both tasks) and those without (No_AD, no deficits in either task).”
- Include mean ± SD or effect size.
Once again, we thank the Reviewer for underscoring this essential aspect and allowing us to improve the quality of our work. Accordingly, we provided the relevant data in our Abstract (please see the Abstract, lines 27-33).
- Weaken causal inferences and avoid absolute conclusions.
We thank the Reviewer for emphasizing this important point, and we agree. Thus, we revised our Abstract as follows (please see the Abstract, lines 35-37):
“These findings indicate that attentional status may affect both adherence to and responsiveness to rehabilitation. This highlights a potentially relevant factor to consider when improving post-stroke interventions.”
Introduction
- The introduction lists a large number of prevalence rates, manifestations, and risks associated with attention deficits (approximately 46–92% of patients, fall risk, mobility/ADL impact, etc.), but lacks summary and critical thinking.
We thank the Reviewer for highlighting this important point, and we agree on the importance of clarifying the functional implications of post-stroke attentional deficits beyond just the epidemiological perspective. Therefore, we revised our manuscript as follows (please see the Introduction section, lines 44-62):
“Among cognitive issues, attention deficits are the most typical in stroke patients [7,8]. Indeed, they affect up to 46–92% of patients during the acute phase, with recovery paths that vary over the following weeks [9,10]. These impairments encompass a wide range, from decreased concentration and distractibility to limited multitasking ability, ultimately impairing daytime functioning and independence [11]. Importantly, beyond their immediate effects, attention deficits are consistently linked to negative outcomes, including a higher risk of falls in older adults and poorer long-term functional recovery [10,12]. This suggests that attentional dysfunction may act less as an isolated symptom and more as a determinant of rehabilitation potential. Indeed, difficulties in maintaining, dividing, or selectively focusing attention have been reported to correlate with decreased mobility, balance, and independence in daily activities up to one year after stroke [9]. Nonetheless, these associations were largely diminished after accounting for baseline functional status, indicating that attentional performance may have a contributory—albeit not primary—role in influencing long-term recovery outcomes, with baseline motor ability emerging as a more robust predictor. Overall, these findings highlight that attentional capacity is not only highly vulnerable after stroke but also plays a crucial role in shaping rehabilitation progress, as attentional status may be a modifiable factor influencing both adherence to and outcomes of rehabilitation.”
References:
- Cramer, S.C., Richards, L.G., Bernhardt, J., Duncan, P., 2023. Cognitive Deficits After Stroke. Stroke 54, 5–9. https://doi.org/10.1161/STROKEAHA.122.041775
- O’Donoghue, M., Leahy, S., Boland, P., Galvin, R., McManus, J., Hayes, S., 2022. Rehabilitation of Cognitive Deficits Poststroke: Systematic Review and Meta-Analysis of Randomized Controlled Trials. Stroke 53, 1700–1710. https://doi.org/10.1161/STROKEAHA.121.034218
- Hyndman, D., Ashburn, A., 2003. People with stroke living in the community: Attention deficits, balance, ADL ability and falls. Disabil Rehabil 25, 817–822. https://doi.org/10.1080/0963828031000122221
- Hyndman, D., Pickering, R.M., Ashburn, A., 2008. The influence of attention deficits on functional recovery post stroke during the first 12 months after discharge from hospital. J Neurol Neurosurg Psychiatry 79, 656–663. https://doi.org/10.1136/jnnp.2007.125609
- Loetscher, T., Potter, K.-J., Wong, D., das Nair, R., 2019. Cognitive rehabilitation for attention deficits following stroke. Cochrane Database Syst Rev 2019, CD002842. https://doi.org/10.1002/14651858.CD002842.pub3
- Chu, Y.-H., Tang, P.-F., Peng, Y.-C., Chen, H.-Y., 2013. Meta-analysis of type and complexity of a secondary task during walking on the prediction of elderly falls. Geriatrics & Gerontology International 13, 289–297. https://doi.org/10.1111/j.1447-0594.2012.00893.x
- The three-step logic of “what existing studies have done—what is missing—what this study addresses” is absent.
We thank the Reviewer for emphasizing this important point and agree. Indeed, we mainly focused on what is currently known in the literature, while we did not adequately explore the missing elements or how our study aims to address these limitations. Therefore, we revised our Introduction in response to the Reviewer's valuable suggestion (please see the Introduction section, lines 102-117):
“Despite this body of evidence, attentional deficits have seldom been investigated as an explicit factor shaping the effectiveness of rehabilitation, particularly within structured interventions such as AOT. Indeed, most of the literature on AOT primarily focused on motor outcomes, often overlooking the dynamic contribution of attention to sustaining engagement and maximizing treatment efficacy [37–39]. Moreover, reliable in-treatment measures of attentional performance have rarely been incorporated, leaving the extent to which attentional deficits hinder participation and responsiveness largely unclear. Therefore, the present pilot study was designed to explore whether attentional performance influences both engagement and motor recovery during a four-week AOT program in subacute stroke patients. Specifically, we aimed to examine the interplay between attentional deficits, patients’ engagement, and their potential to benefit from AOT, while also determining whether AOT could foster improvements in attentional functioning. We hypothesized that attentional performance would significantly influence both adherence to and motor gains from AOT, and that, in turn, the structured demands of sustained concentration and interactive tasks in AOT might further boost participants' attentional capacities.”
References:
- Zhang B, Kan L, Dong A, Zhang J, Bai Z, Xie Y, Liu Q, Peng Y. The effects of action observation training on improving upper limb motor functions in people with stroke: A systematic review and meta-analysis. PLoS One. 2019 Aug 30;14(8):e0221166. doi: 10.1371/journal.pone.0221166. PMID: 31469840; PMCID: PMC6716645.
- Borges LR, Fernandes AB, Oliveira Dos Passos J, Rego IAO, Campos TF. Action observation for upper limb rehabilitation after stroke. Cochrane Database Syst Rev. 2022 Aug 5;8(8):CD011887. doi: 10.1002/14651858.CD011887.pub3. PMID: 35930301; PMCID: PMC9354942.
- Lim H, Jeong CH, Kang YJ, Ku J. Attentional State-Dependent Peripheral Electrical Stimulation During Action Observation Enhances Cortical Activations in Stroke Patients. Cyberpsychol Behav Soc Netw. 2023 Jun;26(6):408-416. doi: 10.1089/cyber.2022.0176. Epub 2023 Apr 20. PMID: 37083413.
- The final sentence of the introduction merely states, “We conducted a pilot study... to examine the role of attention and engagement in motor recovery following AOT,” which is overly broad.
Once again, we thank the Reviewer for underscoring this essential aspect and allowing us to improve the quality of our Introduction. Please note that we have revised this aspect, as already specified in our response to the Reviewer's previous suggestion (please see the Introduction section and the answer to suggestion 2).
- The introduction repeatedly describes the hazards of attention deficits and the importance of engagement, which highly overlaps with the abstract.
We thank the Reviewer for this valuable observation and fully agree on the importance of improving our Introduction. Therefore, in response to this and the earlier comments from the Reviewer, as well as input from other Reviewers, we have significantly revised the Introduction section (please see the revised version of the Introduction). We hope these changes have improved the overall quality of our work by addressing all the suggestions received, for which we are once again sincerely grateful.
Methods and Results
- In such a small sample size, any significant results may be due to chance, especially when reporting “significant predictors” in multiple regression analysis, which has extremely low credibility.
We thank the Reviewer for this comment and and we agree that our results should be tempered in light of the small sample size and the statistical analysis. First of all, we have removed the regression analysis throughout the entire manuscript due to the exploratory nature of the study and the small sample size. Furthermore, we have reformulated the results, adopting a more cautious approach in light of the small sample size, which does not allow us to draw definitive conclusions (please see the revised version of our manuscript).
- The absence of confidence intervals (95% CI) and effect sizes fails to meet the standards of high-quality clinical research.
We thank the Reviewer for this suggestion. We included in Table 1b and Table 2 the 95%CI values. With regard to the effect size, we believe that such a small sample does not allow for obtaining reliable values and we therefore considered it appropriate not to proceed with its calculation. For further information regarding the unreliability of effect size in small samples, please see:
- Button KS, Ioannidis JP, Mokrysz C, et al. Power failure: why small sample size undermines the reliability of neuroscience. Nat Rev Neurosci. 2013;14(5):365-376. doi:10.1038/nrn3475
- Overemphasis on statistical significance.
We agree with the Reviewer on this point and appreciate this suggestion. We have therefore revised our observations based on a statistical analysis with a small sample size (please see the updated version of the Results section).
- In the results section, the authors include explanatory language (e.g., “training became monotonous, leading to decreased accuracy” and “patients with attention deficits felt more fatigued”), which should belong in the discussion rather than the results.
We thank the Reviewer for this suggestion, and we agree. Thus, we have moved these comments to the Discussion section (please see the revised manuscript).
- The legends for Figures 1, 2, and 3 are too general and lack key numerical values.
We thank the Reviewer for this relevant suggestion. We improved the quality of the Figures and their description (please see the revised version of our Figures).
Discussion
- The authors repeatedly emphasize that “attention deficits significantly hinder rehabilitation,” but in fact, only the FMA-UE metric showed significant differences, while other primary metrics did not.
- In a small sample (n=10) + pilot study context, such statements imply overly strong causality.
- It is recommended to soften the tone, e.g., “The results suggest that there may be... which requires validation with a larger sample.”
- While numerous supporting studies are cited, there is insufficient discussion of “inconsistent results or counterexamples.”
- This leads readers to perceive the discussion as overly one-sided, lacking academic balance.
- The discussion section repeats the harm of attention deficits and the importance of engagement (already covered in the introduction), resulting in redundant information.
- The discussion should focus more on “the novel findings and value of this study” and avoid lengthy background reviews.
We sincerely thank the Reviewer for these insightful comments, and we agree. In accordance with this and feedback from the other Reviewers, we have thoroughly revised the Discussion to better highlight the main findings of our study. Specifically, we have minimized digressions about the broader links between attention deficits and secondary outcomes and focused more clearly on their connection to therapeutic results, engagement, and accuracy within the AOT protocol. Additionally, we aimed to offer a more balanced interpretation of the results by considering alternative explanations, avoiding unnecessary overlap with the Introduction, and streamlining the narrative to emphasize the most novel and relevant contributions of our work. Finally, we have tried to soften our statements due to the small sample size and exploratory nature of the study, aiming to suggest ideas for future research rather than making overly definitive claims. We hope these revisions have enhanced the clarity and focus of the Discussion as suggested (please see the revised version of the Discussion).
Conclusion
- However, this study is only a pilot study with n=10, and only the FMA-UE result was significant, while most other functional indicators were not significant. The tone of the conclusion does not align with the actual evidence from the study, appearing overly strong.
- Although the methods section states that the study is a pilot study, the conclusion does not reiterate that “the results are preliminary explorations and should be interpreted with caution.” This may lead readers to mistakenly believe that the results can be directly generalized.
- High-quality journals typically recommend briefly noting at the end of the conclusion that “further validation is needed in large-scale controlled studies.”
- The conclusion does not further elaborate on “what this means for rehabilitation practice.”
We thank the Reviewer for these relevant suggestions about our Conclusion, and we agree. Therefore, we extensively revised our Conclusion section based on the important suggestions provided by the Reviewer to better reflect the main findings and current limitations of our work (please see the Conclusion section):
“This pilot study examined the impact of attentional performance on engagement and motor recovery during an AOT rehabilitation program for subacute stroke patients. Despite the limited sample size, our findings offer preliminary evidence that patients without attentional deficits generally engaged more effectively with the training, demonstrated higher accuracy in interactive tasks, and attained greater improvements in upper limb function—most notably reflected in the FMA-UE, the sole outcome measure exhibiting significant group differences and serving as a pertinent tool for identifying motor restitution following stroke [115]. Conversely, patients with attentional deficits often reported decreased motivation and increased cognitive effort, which may have limited their functional improvements. These results should be interpreted with caution due to the exploratory design and small sample size; however, they underscore the potential importance of attentional resources in influencing responsiveness to cognitively demanding rehabilitative protocols, such as AOT. For clinical practice, this indicates that evaluating and supporting attention may enhance both engagement and treatment outcomes, needed to coherently provide appropriate and tailored rehabilitation interventions. On the other hand, it also suggests that AOT could potentially be used, although the absence of a control group prevents us from confirming this at present, not only to improve motor outcomes in patients with particularly severe motor impairments (i.e., unable to undergo conventional rehabilitation), as already suggested, but also to enhance their attentional performance, which is often impaired in this population [58]. This emphasizes the two-way link between AOT and attentional skills. Larger, well-controlled studies are necessary to verify these findings and determine if adding cognitive screening and personalized support can improve AOT's effectiveness in stroke rehabilitation.”
References:
- Buccino G. Action observation treatment: a novel tool in neurorehabilitation. Philos Trans R Soc Lond B Biol Sci. 2014 Apr 28;369(1644):20130185. doi: 10.1098/rstb.2013.0185. PMID: 24778380; PMCID: PMC4006186.
- Kwakkel G, Stinear C, Essers B, Munoz-Novoa M, Branscheidt M, Cabanas-Valdés R, Lakičević S, Lampropoulou S, Luft AR, Marque P, Moore SA, Solomon JM, Swinnen E, Turolla A, Alt Murphy M, Verheyden G. Motor rehabilitation after stroke: European Stroke Organisation (ESO) consensus-based definition and guiding framework. Eur Stroke J. 2023 Dec;8(4):880-894. doi: 10.1177/23969873231191304. Epub 2023 Aug 7. PMID: 37548025; PMCID: PMC10683740.
Once again, we thank the Reviewer for all the valuable suggestions provided.
Reviewer 3 Report
Comments and Suggestions for Authors<Introduction>
1. Line 50-53: This sentence implies a causal relationship between attention deficits at discharge and poor long-term functional outcomes, which is an overstatement based on Reference 9. In fact, Hyndman et al. (2008) found only correlational relationships, and these largely disappeared after adjusting for baseline functional status. Their regression analysis clearly showed that baseline motor ability—not attentional performance—was the main predictor of recovery at 12 months. Please revise this sentence to reflect the limited and non-causal nature of this association. Misrepresenting correlational evidence as causal undermines the credibility of your theoretical framing.
2. Lines 65–74: The introduction of Action Observation Training (AOT) and the mirror neuron system (MNS) is abrupt and lacks a logical bridge from the preceding discussion. The authors fail to articulate why AOT is particularly relevant to the current study’s focus on attention and engagement. No rationale is given for choosing AOT over other motor rehabilitation approaches, and the supposed mechanistic suitability of MNS is merely implied, not demonstrated. The section reads more like an isolated literature summary than a reasoned justification. This weakens the theoretical coherence of the introduction.
3. Lines 75–78: The research gap and rationale for this pilot study are introduced far too late and with insufficient clarity. The manuscript fails to articulate a specific research question or hypothesis, which is essential—even in a pilot design. The transition from the general discussion to the specific aims of the study lacks narrative flow and conceptual sharpness. As written, the paragraph does not convincingly explain what is unknown, why it matters, and how this study addresses that gap.
<Methods>
1. The manuscript introduces a self-developed 8-item “AOT Engagement Questionnaire,” yet it lacks any description of its development process, psychometric validation, or reliability. Without such information, the interpretability and scientific credibility of the engagement scores are substantially limited, raising concerns about the robustness of the outcome measures.
2. The manuscript classifies participants into AD and No_AD groups based on TAP scores, yet it does not clearly define the cutoff criteria, reference values, or thresholds used for this classification. The lack of justification or citation for these grouping decisions raises concerns about the objectivity and reproducibility of the subgroup analysis. Explicitly stating the cutoff scores and referencing established norms or literature is necessary to ensure methodological transparency.
3. Adherence to the intervention was high, but it fails to provide a clear definition or threshold for what constitutes "high adherence" (e.g., percentage of sessions completed). Without such criteria, the claim lacks transparency and limits the interpretability of intervention fidelity.
<Results>
1. The study conducts multiple regression and correlation analyses despite a very small sample (n = 10), presenting statistically significant predictors. However, the risk of overfitting and Type I error is high. For example, the reported R² = .616 with three predictors may be unstable. These findings should be interpreted as preliminary trends rather than conclusive evidence.
2. The self-developed 8-item engagement questionnaire was used as a key outcome measure, yet no information is provided on its psychometric validation or reliability. Given its central role in the analysis, the absence of validation data undermines the interpretability of the results. The tool should be explicitly described as an exploratory, non-validated measure.
3. The No_AD group consisted predominantly of left-handed individuals (75%), which may have introduced a confounding effect, especially in a visuomotor training context using unilateral video stimuli. This imbalance could have influenced performance outcomes. A post hoc analysis or at minimum, an acknowledgment of this limitation, is warranted.
4. The lack of a control group makes it difficult to differentiate the effects of the AOT intervention from spontaneous motor recovery typically seen in the subacute phase of stroke. Although this is acknowledged in the Discussion, parts of the Results section interpret improvements as effects of the intervention. These claims should be more cautiously framed.
<Discussion>
1.The discussion occasionally implies causal relationships despite the study’s small sample size and non-randomized design. Findings should be presented as associations or preliminary trends rather than confirmed effects.
2. The interpretation of cognitive reserve (CRIq) appears overstated, suggesting a mechanistic role without longitudinal or mediation analysis. Its relationship with outcomes should be framed as correlational.
3. Several uncontrolled confounding factors—such as neglect, mood symptoms, and right-hemisphere lesions—were more prevalent in the attention-deficit group. These differences may account for group effects and should be acknowledged.
4. The discussion does not sufficiently emphasize the limited generalizability due to the small sample size. Stronger caution is needed when drawing clinical implications.
5. Despite identifying attentional deficits as a barrier to engagement, the authors do not propose how AOT could be adapted to accommodate such patients. Future research directions should include attention-supportive modifications.
Author Response
<Introduction>
- Line 50-53: This sentence implies a causal relationship between attention deficits at discharge and poor long-term functional outcomes, which is an overstatement based on Reference 9. In fact, Hyndman et al. (2008) found only correlational relationships, and these largely disappeared after adjusting for baseline functional status. Their regression analysis clearly showed that baseline motor ability—not attentional performance—was the main predictor of recovery at 12 months. Please revise this sentence to reflect the limited and non-causal nature of this association. Misrepresenting correlational evidence as causal undermines the credibility of your theoretical framing.
We would like to thank the Reviewer for taking the time to evaluate our work and for providing helpful feedback. We apologise for this inaccuracy and agree on the importance of reviewing and correcting this aspect. Therefore, also considering input from other Reviewers as well, we have revised the Introduction and, regarding the passage mentioned, we have made the following changes (please see the Introduction section, lines 53-59):
“Indeed, difficulties in maintaining, dividing, or selectively focusing attention have been reported to correlate with decreased mobility, balance, and independence in daily activities up to one year after stroke [9]. Nonetheless, these associations were largely diminished after accounting for baseline functional status, indicating that attentional performance may have a contributory—albeit not primary—role in influencing long-term recovery outcomes, with baseline motor ability emerging as a more robust predictor.”
Reference:
Hyndman, D., Pickering, R.M., Ashburn, A., 2008. The influence of attention deficits on functional recovery post stroke during the first 12 months after discharge from hospital. J Neurol Neurosurg Psychiatry 79, 656–663. https://doi.org/10.1136/jnnp.2007.125609
- Lines 65–74: The introduction of Action Observation Training (AOT) and the mirror neuron system (MNS) is abrupt and lacks a logical bridge from the preceding discussion. The authors fail to articulate why AOT is particularly relevant to the current study’s focus on attention and engagement. No rationale is given for choosing AOT over other motor rehabilitation approaches, and the supposed mechanistic suitability of MNS is merely implied, not demonstrated. The section reads more like an isolated literature summary than a reasoned justification. This weakens the theoretical coherence of the introduction.
We sincerely thank the Reviewer for this relevant suggestion, and we agree on the relevance of clarifying this essential point. Thus, also considering input from other Reviewers, we revised our Introduction as follows (please see the Introduction section, lines 74-101):
“It is essential to acknowledge that patients with severe motor deficits often present with concomitant attentional impairments, as their lesions frequently extend to regions involved in both cognitive and motor functions [22,23]. Furthermore, their level of impairment generally hinders them from engaging in standard neurorehabilitation protocols that demand at least some active participation [24,25]. Since these patients generally have the poorest long-term outcomes, there is an urgent need for strategies applicable to individuals with severe motor impairments that can also address the cognitive dimensions of recovery, with attention being a particularly important target [26]. Recently, new rehabilitation strategies have been proposed to increase upper limb motor recovery after stroke, including action observation training (AOT), a mirror neuron system (MNS)-based approach that typically involves observing and performing goal-directed daily actions [27–30]. Specifically, it induces targeted motor facilitation in the corticospinal system, increasing the excitability of the injured sensorimotor system in the primary motor cortex and promoting brain reorganization by activating central representations of actions through the MNS [28]. It was also consistently demonstrated that AOT prevents corticomotor depression induced by immobilization during limb inactivity [31]. Importantly, since AOT depends on activating the MNS and related frontoparietal networks, it is closely connected with attentional control systems [32]. Therefore, successful engagement in AOT requires focusing attention to observe, process, and mentally simulate the watched motor actions [33]. Moreover, directing visual attention explicitly during AOT facilitates corticospinal excitability and may accelerate motor relearning through observation [34]. Attention deficits may limit the ability to properly encode and integrate observed movements, thereby reducing the effectiveness of AOT [35]. On the other hand, structured AOT protocols might also help train attention skills, as patients must repeatedly stay focused and selectively process relevant motor cues [36]. Thus, attention is both a necessary prerequisite and a potential focus in AOT, and understanding this relationship is essential for optimizing patient selection and therapeutic results.”
References:
- D'Imperio D, Romeo Z, Maistrello L, Durgoni E, Della Pietà C, De Filippo De Grazia M, Meneghello F, Turolla A, Zorzi M. Sensorimotor, Attentional, and Neuroanatomical Predictors of Upper Limb Motor Deficits and Rehabilitation Outcome after Stroke. Neural Plast. 2021 Apr 1;2021:8845685. doi: 10.1155/2021/8845685. PMID: 33868400; PMCID: PMC8035034.
- Evangelista GG, Egger P, Brügger J, Beanato E, Koch PJ, Ceroni M, Fleury L, Cadic-Melchior A, Meyer NH, Rodríguez DL, Girard G, Léger B, Turlan JL, Mühl A, Vuadens P, Adolphsen J, Jagella CE, Constantin C, Alvarez V, San Millán D, Bonvin C, Morishita T, Wessel MJ, Van De Ville D, Hummel FC. Differential Impact of Brain Network Efficiency on Poststroke Motor and Attentional Deficits. Stroke. 2023 Apr;54(4):955-963. doi: 10.1161/STROKEAHA.122.040001. Epub 2023 Feb 27. PMID: 36846963; PMCID: PMC10662579.
- Stolwyk RJ, Mihaljcic T, Wong DK, Chapman JE, Rogers JM. Poststroke Cognitive Impairment Negatively Impacts Activity and Participation Outcomes: A Systematic Review and Meta-Analysis. Stroke. 2021 Jan;52(2):748-760. doi: 10.1161/STROKEAHA.120.032215. Epub 2021 Jan 25. PMID: 33493048.
- James J, McGlinchey MP. How active are stroke patients in physiotherapy sessions and is this associated with stroke severity? Disabil Rehabil. 2022 Aug;44(16):4408-4414. doi: 10.1080/09638288.2021.1907459. Epub 2021 Apr 1. PMID: 33794718.
- Winstein CJ, Stein J, Arena R, Bates B, Cherney LR, Cramer SC, Deruyter F, Eng JJ, Fisher B, Harvey RL, Lang CE, MacKay-Lyons M, Ottenbacher KJ, Pugh S, Reeves MJ, Richards LG, Stiers W, Zorowitz RD; American Heart Association Stroke Council, Council on Cardiovascular and Stroke Nursing, Council on Clinical Cardiology, and Council on Quality of Care and Outcomes Research. Guidelines for Adult Stroke Rehabilitation and Recovery: A Guideline for Healthcare Professionals From the American Heart Association/American Stroke Association. Stroke. 2016 Jun;47(6):e98-e169. doi: 10.1161/STR.0000000000000098. Epub 2016 May 4. Erratum in: Stroke. 2017 Feb;48(2):e78. doi: 10.1161/STR.0000000000000120. Erratum in: Stroke. 2017 Dec;48(12):e369. doi: 10.1161/STR.0000000000000156. PMID: 27145936.
- Antonioni, A., Raho, E.M., Straudi, S., Granieri, E., Koch, G., Fadiga, L., 2024b. The cerebellum and the Mirror Neuron System: a matter of inhibition? From neurophysiological evidence to neuromodulatory implications. A narrative review. Neuroscience & Biobehavioral Reviews 105830. https://doi.org/10.1016/j.neubiorev.2024.105830
- Antonioni, A., Galluccio, M., Baroni, A., Fregna, G., Pozzo, T., Koch, G., Manfredini, F., Fadiga, L., Malerba, P., Straudi, S., 2024a. Event-related desynchronization during action observation is an early predictor of recovery in subcortical stroke: An EEG study. Ann Phys Rehabil Med 67, 101817. https://doi.org/10.1016/j.rehab.2024.101817
- Sarasso, E., Gemma, M., Agosta, F., Filippi, M., Gatti, R., 2015. Action observation training to improve motor function recovery: a systematic review. Arch Physiother 5, 14. https://doi.org/10.1186/s40945-015-0013-x
- Zhang, J.J.Q., Fong, K.N.K., Welage, N., Liu, K.P.Y., 2018. The Activation of the Mirror Neuron System during Action Observation and Action Execution with Mirror Visual Feedback in Stroke: A Systematic Review. Neural Plast 2018, 2321045. https://doi.org/10.1155/2018/2321045
- Nummenmaa L, Smirnov D, Lahnakoski JM, Glerean E, Jääskeläinen IP, Sams M, Hari R. Mental action simulation synchronizes action-observation circuits across individuals. J Neurosci. 2014 Jan 15;34(3):748-57. doi: 10.1523/JNEUROSCI.0352-13.2014. PMID: 24431433; PMCID: PMC3891955.
- Kemmerer D. What modulates the Mirror Neuron System during action observation?: Multiple factors involving the action, the actor, the observer, the relationship between actor and observer, and the context. Prog Neurobiol. 2021 Oct;205:102128. doi: 10.1016/j.pneurobio.2021.102128. Epub 2021 Jul 31. PMID: 34343630.
- Sánchez Silverio V, Abuín Porras V, Rodríguez Costa I, Cleland JA, Villafañe JH. Effects of action observation training on the walking ability of patients post stroke: a systematic review. Disabil Rehabil. 2022 Dec;44(24):7339-7348. doi: 10.1080/09638288.2021.1989502. Epub 2021 Oct 13. PMID: 34644226.
- Caligiore D, Mustile M, Fineschi A, Romano L, Piras F, Assogna F, Pontieri FE, Spalletta G, Baldassarre G. Action Observation With Dual Task for Improving Cognitive Abilities in Parkinson's Disease: A Pilot Study. Front Syst Neurosci. 2019 Feb 11;13:7. doi: 10.3389/fnsys.2019.00007. PMID: 30804762; PMCID: PMC6378302.
- D’Innocenzo, G., Gonzalez, C.C., Nowicky, A.V., Williams, A.M., Bishop, D.T., 2017. Motor resonance during action observation is gaze-contingent: A TMS study. Neuropsychologia 103, 77–86. https://doi.org/10.1016/j.neuropsychologia.2017.07.017
- Lines 75–78: The research gap and rationale for this pilot study are introduced far too late and with insufficient clarity. The manuscript fails to articulate a specific research question or hypothesis, which is essential—even in a pilot design. The transition from the general discussion to the specific aims of the study lacks narrative flow and conceptual sharpness. As written, the paragraph does not convincingly explain what is unknown, why it matters, and how this study addresses that gap.
Once again, we thank the Reviewer for highlighting this crucial point, and we agree. Indeed, following the recommendations provided by other Reviewers, we revised the entire Introduction. Specifically, about the rationale and aims of our study, we revised as follows (please see the Introduction section, lines 102-117):
“Despite this body of evidence, attentional deficits have seldom been investigated as an explicit factor shaping the effectiveness of rehabilitation, particularly within structured interventions such as AOT. Indeed, most of the literature on AOT primarily focused on motor outcomes, often overlooking the dynamic contribution of attention to sustaining engagement and maximizing treatment efficacy [37–39]. Moreover, reliable in-treatment measures of attentional performance have rarely been incorporated, leaving the extent to which attentional deficits hinder participation and responsiveness largely unclear. Therefore, the present pilot study was designed to explore whether attentional performance influences both engagement and motor recovery during a four-week AOT program in subacute stroke patients. Specifically, we aimed to examine the interplay between attentional deficits, patients’ engagement, and their potential to benefit from AOT, while also determining whether AOT could foster improvements in attentional functioning. We hypothesized that attentional performance would significantly influence both adherence to and motor gains from AOT, and that, in turn, the structured demands of sustained concentration and interactive tasks in AOT might further boost participants' attentional capacities.”
References:
- Borges LR, Fernandes AB, Oliveira Dos Passos J, Rego IAO, Campos TF. Action observation for upper limb rehabilitation after stroke. Cochrane Database Syst Rev. 2022 Aug 5;8(8):CD011887. doi: 10.1002/14651858.CD011887.pub3. PMID: 35930301; PMCID: PMC9354942.
- Zhang B, Kan L, Dong A, Zhang J, Bai Z, Xie Y, Liu Q, Peng Y. The effects of action observation training on improving upper limb motor functions in people with stroke: A systematic review and meta-analysis. PLoS One. 2019 Aug 30;14(8):e0221166. doi: 10.1371/journal.pone.0221166. PMID: 31469840; PMCID: PMC6716645.
- Lim H, Jeong CH, Kang YJ, Ku J. Attentional State-Dependent Peripheral Electrical Stimulation During Action Observation Enhances Cortical Activations in Stroke Patients. Cyberpsychol Behav Soc Netw. 2023 Jun;26(6):408-416. doi: 10.1089/cyber.2022.0176. Epub 2023 Apr 20. PMID: 37083413.
<Methods>
- The manuscript introduces a self-developed 8-item “AOT Engagement Questionnaire,” yet it lacks any description of its development process, psychometric validation, or reliability. Without such information, the interpretability and scientific credibility of the engagement scores are substantially limited, raising concerns about the robustness of the outcome measures.
We agree with the Reviewer on the importance of this point. Therefore, in the Methods section, we have added the process behind the creation of the questionnaire (please see lines 175-177):
“Items were generated based on a review of engagement constructs in rehabilitation and refined through discussions with the clinical research team.”
However, agreeing with the Reviewer about the limited interpretability of the tool without scientific validation, we have listed the use of an unvalidated questionnaire as a limitation of the manuscript (please see the Discussion section, lines 530-535):
“Another limitation concerns the AOT engagement questionnaire, which was specifically developed for this study. Although designed to capture satisfaction, fatigue, and motivation, the instrument has not undergone formal psychometric validation, which restricts the interpretability of the engagement results. Future research should address this gap by evaluating the reliability and validity of the scale in larger and more diverse cohorts.”
- The manuscript classifies participants into AD and No_AD groups based on TAP scores, yet it does not clearly define the cutoff criteria, reference values, or thresholds used for this classification. The lack of justification or citation for these grouping decisions raises concerns about the objectivity and reproducibility of the subgroup analysis. Explicitly stating the cutoff scores and referencing established norms or literature is necessary to ensure methodological transparency.
We thank the Reviewer for highlighting this important point, and we entirely agree on the importance of clarifying it. As outlined in the Methods section, patients were categorized into groups with or without attentional deficits based on their performance on two subtests of the Test of Attentional Performance (TAP), specifically divided attention and Go/No-Go. Specifically, patients were classified as “without deficits” only if their performance in both subtests fell within the normal range, while an impairment in even one of the two subtests led to classification in the “with attentional deficits” group. This approach was chosen to increase the accuracy of classification for patients without deficits, as previously suggested by other studies. To clarify this important point, we added the following to our manuscript (please see subsection 2.1, lines 156-162):
“Consistent with the literature, impairment on the TAP divided attention and Go/No-Go subtests was defined as performance below the 5th percentile of age- and education-adjusted normative data (collected by the Neuropsychology Service of the Medical Rehabilitation Unit of the Ferrara University Hospital to ensure alignment with the socio-cultural context of the study population), based on reaction times and error rates (omissions and commissions) [47–49]. Due to the specific focus of this work, analyses were conducted on reaction times.”
We hope this clarification addresses the Reviewer’s concern and, at the same time, improves the overall quality and transparency of our work. We are sincerely grateful to the Reviewer for pointing out this important aspect.
References:
- Schmidt SL, Boechat YEM, Schmidt GJ, Nicaretta D, van Duinkerken E, Schmidt JJ. Clinical Utility of a Reaction-Time Attention Task in the Evaluation of Cognitive Impairment in Elderly with High Educational Disparity. J Alzheimers Dis. 2021;81(2):691-697. doi: 10.3233/JAD-210151. PMID: 33814451.
- Schmidt GJ, Boechat YEM, van Duinkerken E, Schmidt JJ, Moreira TB, Nicaretta DH, Schmidt SL. Detection of Cognitive Dysfunction in Elderly with a Low Educational Level Using a Reaction-Time Attention Task. J Alzheimers Dis. 2020;78(3):1197-1205. doi: 10.3233/JAD-200881. PMID: 33136095.
- Schumacher R, Halai AD, Lambon Ralph MA. Attention to attention in aphasia - elucidating impairment patterns, modality differences and neural correlates. Neuropsychologia. 2022 Dec 15;177:108413. doi: 10.1016/j.neuropsychologia.2022.108413. Epub 2022 Nov 3. PMID: 36336090; PMCID: PMC7614452.
- Adherence to the intervention was high, but it fails to provide a clear definition or threshold for what constitutes "high adherence" (e.g., percentage of sessions completed). Without such criteria, the claim lacks transparency and limits the interpretability of intervention fidelity.
We agree with the Reviewer about this point. In our idea, we assumed that all subjects completed the treatment programme and therefore modified the sentence in the Results section (please see line 307):
“All participants in both groups completed the training program.”
Furthermore, to improve reporting accuracy, we specified the number of sessions actually completed by participants, also following another Reviewer's suggestion (please see lines 266-270):
“The features of training, expressed as the mean values (range), were as follows: 6 sessions per week (3–10), 16.2 days (1–40) of treatment duration, 12.4 minutes (5–30) per session, and 16.9 minutes of observed action performance (5–36). Therefore, compared to the planned protocol, the treatment was given at lower doses, which is common in real-world clinical practice [38].”
We hope that this underscores the absence of dropouts from our study population, as well as the real-world challenges involved in conducting studies like this.
Reference:
Borges, L.R.; Fernandes, A.B.; Oliveira Dos Passos, J.; Rego, I.A.O.; Campos, T.F. Action Observation for Upper Limb Rehabilitation after Stroke. Cochrane Database Syst Rev 2022, 8, CD011887, doi:10.1002/14651858.CD011887.pub3.
<Results>
- The study conducts multiple regression and correlation analyses despite a very small sample (n = 10), presenting statistically significant predictors. However, the risk of overfitting and Type I error is high. For example, the reported R² = .616 with three predictors may be unstable. These findings should be interpreted as preliminary trends rather than conclusive evidence.
We thank the Reviewer for highlighting this significant point and agree. In fact, we endeavored to utilize these analyses to investigate the complex relationship between the variables of interest in the most appropriate manner, to propose potential evidence for future research. Nonetheless, we agree that, considering the sample size and the exploratory nature of the study, such analyses are inappropriate and the results may be misleading. Therefore, also following the suggestion from other Reviewers, we have removed references to regressions throughout the entire manuscript (please see the revised version). We appreciate the Reviewer for helping us improve the scientific accuracy of our work.
- The self-developed 8-item engagement questionnaire was used as a key outcome measure, yet no information is provided on its psychometric validation or reliability. Given its central role in the analysis, the absence of validation data undermines the interpretability of the results. The tool should be explicitly described as an exploratory, non-validated measure.
We agree with the Reviewer on the importance of this point. In the Methods section, we have outlined the process of developing the questionnaire and its exploratory approach (please see lines 175-177):
“Items were generated based on a review of engagement constructs in rehabilitation and refined through discussions with the clinical research team.”
However, agreeing with the Reviewer about the limited interpretability of the tool without scientific validation, we have listed the use of an unvalidated questionnaire as a limitation of the manuscript (please see the Discussion section, lines 530-535):
“Another limitation concerns the AOT engagement questionnaire, which was specifically developed for this study. Although designed to capture satisfaction, fatigue, and motivation, the instrument has not undergone formal psychometric validation, which restricts the interpretability of the engagement results. Future research should address this gap by evaluating the reliability and validity of the scale in larger and more diverse cohorts.”
- The No_AD group consisted predominantly of left-handed individuals (75%), which may have introduced a confounding effect, especially in a visuomotor training context using unilateral video stimuli. This imbalance could have influenced performance outcomes. A post hoc analysis or at minimum, an acknowledgment of this limitation, is warranted.
We thank the Reviewer for this insightful observation. We fully acknowledge that the predominance of left-handed individuals in the No_AD group (75%) may represent a potential source of bias, particularly in the context of visuomotor training based on unilateral video stimuli. Unfortunately, given the small sample size of this pilot study, a reliable post hoc analysis to disentangle the possible influence of handedness on performance outcomes is not feasible. Nonetheless, we revised the Discussion to clearly acknowledge this imbalance as a limitation and to mention that handedness may have interacted with attentional status in influencing training engagement and motor improvements (please see the Discussion section, lines 498-501):
“Another limitation of this pilot study is the imbalance in handedness between groups, as the No_AD group included a predominance of left-handed individuals. This may have introduced a potential confounding effect, especially in a visuomotor training context involving unilateral video stimuli [104,105].”
We thank the Reviewer for highlighting this essential point.
References:
- Crotti M, Koschutnig K, Wriessnegger SC. Handedness impacts the neural correlates of kinesthetic motor imagery and execution: A FMRI study. J Neurosci Res. 2022 Mar;100(3):798-826. doi: 10.1002/jnr.25003. Epub 2022 Jan 3. PMID: 34981561; PMCID: PMC9303560.
- Kirby KM, Pillai SR, Carmichael OT, Van Gemmert AWA. Brain functional differences in visuo-motor task adaptation between dominant and non-dominant hand training. Exp Brain Res. 2019 Dec;237(12):3109-3121. doi: 10.1007/s00221-019-05653-5. Epub 2019 Sep 21. PMID: 31542802.
- The lack of a control group makes it difficult to differentiate the effects of the AOT intervention from spontaneous motor recovery typically seen in the subacute phase of stroke. Although this is acknowledged in the Discussion, parts of the Results section interpret improvements as effects of the intervention. These claims should be more cautiously framed.
We thank the Reviewer for pointing out this essential aspect. We agree that the absence of a control group limits the ability to attribute improvements solely to the AOT intervention, as spontaneous recovery is expected in the subacute phase. We have revised the Results, Discussion, and Conclusions sections to ensure that statements about improvements are framed more cautiously and to emphasize that findings should be interpreted as preliminary and exploratory (please see the revised version of the manuscript).
<Discussion>
1.The discussion occasionally implies causal relationships despite the study’s small sample size and non-randomized design. Findings should be presented as associations or preliminary trends rather than confirmed effects.
We sincerely thank the Reviewer for highlighting this essential point, and we agree. In accordance with this and feedback from the other Reviewers, we have thoroughly revised the Discussion to better highlight the main findings of our study. Additionally, we aimed to offer a more balanced interpretation of the results by considering alternative explanations, avoiding unnecessary overlap with the Introduction, and streamlining the narrative to emphasize the most novel and relevant contributions of our work. Finally, we have tried to soften our statements due to the small sample size and exploratory nature of the study, aiming to suggest ideas for future research rather than making overly definitive claims. We hope these revisions have enhanced the clarity and focus of the Discussion as suggested (please see the revised version of the Discussion).
- The interpretation of cognitive reserve (CRIq) appears overstated, suggesting a mechanistic role without longitudinal or mediation analysis. Its relationship with outcomes should be framed as correlational.
We would like to thank the Reviewer for this relevant comment, and we agree. The effects of cognitive reserve and their limitations on the outcomes investigated have been revised in both the Results and Discussion sections (please see the revised manuscript).
- Several uncontrolled confounding factors—such as neglect, mood symptoms, and right-hemisphere lesions—were more prevalent in the attention-deficit group. These differences may account for group effects and should be acknowledged.
We thank the Reviewer for this important observation. We agree that neglect, mood symptoms, and right-hemisphere lesions were more common in the attention-deficit group and could have affected the observed group differences. Therefore, we revised the Discussion section to clearly acknowledge this limitation and highlight the importance of future studies with larger samples that enable statistical control of these variables (please see the Discussion, lines 501-509):
“Furthermore, in our sample, neglect, mood symptoms, and right-hemisphere lesions were more prevalent in the AD group. These factors are known to influence both cognitive and motor outcomes after stroke and may have contributed to the observed group differences. Specifically, spatial neglect adversely affects rehabilitation outcomes, leading to slower functional gains, higher risk of falls, and longer hospital stays [106]. Additionally, common post-stroke mood disorders like depression, apathy, and anxiety negatively affect both cognitive and functional recovery, contributing to increased disability and poorer outcomes [107].”
References:
- Embrechts, E.; Loureiro-Chaves, R.; Nijboer, T.C.W.; Lafosse, C.; Truijen, S.; Saeys, W. The Association of Personal Neglect with Motor, Activities of Daily Living, and Participation Outcomes after Stroke: A Systematic Review. Arch Clin Neuropsychol 2024, 39, 249–264, doi:10.1093/arclin/acad063.
- Hama, S.; Yoshimura, K.; Yanagawa, A.; Shimonaga, K.; Furui, A.; Soh, Z.; Nishino, S.; Hirano, H.; Yamawaki, S.; Tsuji, T. Relationships between Motor and Cognitive Functions and Subsequent Post-Stroke Mood Disorders Revealed by Machine Learning Analysis. Sci Rep 2020, 10, 19571, doi:10.1038/s41598-020-76429-z.
- The discussion does not sufficiently emphasize the limited generalizability due to the small sample size. Stronger caution is needed when drawing clinical implications.
Once again, we thank the Reviewer for pointing out this crucial point, and we agree. As already mentioned in our response to the Reviewer’s previous comment, we have thoroughly revised the entire Discussion to tone down overly strong statements, considering the exploratory nature of the study and the small sample size. We have also clarified our interpretation of the findings in the Conclusions by explicitly emphasizing that the generalizability of the results should be approached with caution (please see the revised version of the Discussion and Conclusions). In doing so, we aimed to highlight the significance of our preliminary findings while, as the Reviewer correctly suggested, avoiding definitive claims and instead prompting readers to interpret the results carefully. We sincerely thank the Reviewer for this helpful feedback, which enabled us to present our work more accurately and transparently.
- Despite identifying attentional deficits as a barrier to engagement, the authors do not propose how AOT could be adapted to accommodate such patients. Future research directions should include attention-supportive modifications.
We thank the Reviewer for this clinically relevant suggestion. Concerning the impact of attention in modulating subjects’ engagement in rehabilitation treatment, as addressed in our work, the need to adapt therapeutic interventions according to individuals’ cognitive status is specified more in the Discussion and Conclusions section (for instance, please see lines 385-387 and 565-568):
“Thus, in clinical practice, tailored AOT rehabilitation interventions specifically adapted to cognitive and attentional abilities appear to be relevant in driving subjects’ engagement and motor recovery.”
“For clinical practice, this indicates that evaluating and supporting attention may enhance both engagement and treatment outcomes, needed to coherently provide appropriate and tailored rehabilitation interventions.”
Reviewer 4 Report
Comments and Suggestions for Authors
Title: Action Observation Training for Upper Limb Stroke Rehabilitation: A Pilot Study on the Role of Attention and Patient Engagement.
Strengths of the Article:
The manuscript addresses an important and underexplored aspect of stroke rehabilitation: the interaction between attentional deficits, patient engagement, and responsiveness to action observation training (AOT). The pilot study design, while limited by sample size (n=10), provides meaningful preliminary evidence that attentional status influences motor recovery outcomes. The paper is well-structured, the methods are described in sufficient detail, and the discussion contextualizes the results, considering existing literature.
The study addresses a clinically meaningful question and presents promising pilot data. However, stronger acknowledgment of methodological constraints (sample size, absence of a control group, and exploratory engagement measure) and clearer justification of group classifications are needed before the manuscript can be considered for publication.
Major Comments
- The pilot nature of the study is acknowledged, but the small cohort (n=10, with further division into AD and No-AD subgroups) significantly limits statistical power.
- The authors should more explicitly emphasize that conclusions are preliminary and should be validated in larger randomized controlled trials.
- Without a conventional therapy or sham intervention group, it is difficult to isolate the effect of AOT from spontaneous recovery in the subacute stroke phase. While the authors discuss this, it deserves stronger emphasis in both results interpretation and limitations.
- The classification into AD vs. No-AD groups is central to the manuscript. However, the operational definition relies on TAP subtests only. Please clarify the cutoff criteria more explicitly and discuss whether this captures the heterogeneity of attentional impairments (sustained, selective, divided).
- The self-developed 8-item AOT engagement questionnaire is an interesting tool, but its validation status remains unclear.
- The discussion highlights attentional deficits as predictors of reduced engagement and motor gains, which is well supported. However, alternative explanations (e.g., mood disturbances, cognitive reserve differences) deserve more nuanced integration, since they co-varied with attentional performance.
Minor Comments
- Abstract: Consider reporting exact subgroup sizes (AD = 6, No-AD = 4) to provide context for readers.
- Methods: Please clarify whether therapists were blinded to patients’ attentional classification when delivering AOT. Lack of blinding could bias patient-therapist interaction.
- Figures: Figures 2–4 are informative but could be improved by including sample sizes (N=) in the legends and ensuring all abbreviations are defined within each caption.
- Tables: In Table 2, reporting effect sizes (e.g., Cohen’s d) in addition to p-values would strengthen interpretation, given the small sample.
- Language: Minor grammatical inconsistencies (e.g., “patients with attention deficits reported a minor sense of accomplishment” → “patients…reported a reduced sense of accomplishment”).
Minor grammatical inconsistencies (e.g., “patients with attention deficits reported a minor sense of accomplishment” → “patients…reported a reduced sense of accomplishment”).
Author Response
Strengths of the Article:
The manuscript addresses an important and underexplored aspect of stroke rehabilitation: the interaction between attentional deficits, patient engagement, and responsiveness to action observation training (AOT). The pilot study design, while limited by sample size (n=10), provides meaningful preliminary evidence that attentional status influences motor recovery outcomes. The paper is well-structured, the methods are described in sufficient detail, and the discussion contextualizes the results, considering existing literature.
The study addresses a clinically meaningful question and presents promising pilot data. However, stronger acknowledgment of methodological constraints (sample size, absence of a control group, and exploratory engagement measure) and clearer justification of group classifications are needed before the manuscript can be considered for publication.
Major Comments
The pilot nature of the study is acknowledged, but the small cohort (n=10, with further division into AD and No-AD subgroups) significantly limits statistical power.
The authors should more explicitly emphasize that conclusions are preliminary and should be validated in larger randomized controlled trials.
We would like to thank the Reviewer for taking the time to evaluate our work and for providing helpful feedback to improve its quality. We sincerely thank the Reviewer for highlighting this essential point, and we agree. In accordance with this and feedback from the other Reviewers, we have thoroughly revised the entire Discussion to tone down overly strong statements, considering the exploratory nature of the study and the small sample size. We have also clarified our interpretation of the findings in the Conclusions by explicitly emphasizing that the generalizability of the results should be approached with caution (please see the revised version of the Discussion and Conclusions). In doing so, we aimed to highlight the significance of our preliminary findings while, as the Reviewer correctly suggested, avoiding definitive claims and instead prompting readers to interpret the results carefully. We sincerely thank the Reviewer for this helpful feedback, which enabled us to present our work more accurately and transparently.
Without a conventional therapy or sham intervention group, it is difficult to isolate the effect of AOT from spontaneous recovery in the subacute stroke phase. While the authors discuss this, it deserves stronger emphasis in both results interpretation and limitations.
Once again, we thank the Reviewer for this relevant suggestion, and we agree. Please note that, in addition to thoroughly revising the entire manuscript—especially the Discussion and Conclusion sections (as explained in the previous comment)—to better highlight the study's limitations, including the lack of a control group, we have also explicitly added the following statement at the beginning of the Discussion (please refer to the revised versions of the Discussion, lines 378-380):
“Nevertheless, the absence of a control group receiving an alternative treatment hinders our ability to attribute the observed differences to the AOT protocol definitively.”
This was done to prompt readers to interpret the findings carefully and with proper awareness. We thank the Reviewer for emphasizing this important point.
The classification into AD vs. No-AD groups is central to the manuscript. However, the operational definition relies on TAP subtests only. Please clarify the cutoff criteria more explicitly and discuss whether this captures the heterogeneity of attentional impairments (sustained, selective, divided).
We thank the Reviewer for highlighting this important point, and we entirely agree on the importance of clarifying it. As outlined in the Methods section, patients were categorized into groups with or without attentional deficits based on their performance on two subtests of the Test of Attentional Performance (TAP), specifically divided attention and Go/No-Go. Specifically, patients were classified as “without deficits” only if their performance in both subtests fell within the normal range, while an impairment in even one of the two subtests led to classification in the “with attentional deficits” group. This approach was chosen to increase the accuracy of classification for patients without deficits, as previously suggested by other studies. To clarify this important point, we added the following to our manuscript (please see subsection 2.1, lines 156-162):
“Consistent with the literature, impairment on the TAP divided attention and Go/No-Go subtests was defined as performance below the 5th percentile of age- and education-adjusted normative data (collected by the Neuropsychology Service of the Medical Rehabilitation Unit of the Ferrara University Hospital to ensure alignment with the socio-cultural context of the study population), based on reaction times and error rates (omissions and commissions) [47–49]. Due to the specific focus of this work, analyses were conducted on reaction times.”
References:
- Schmidt SL, Boechat YEM, Schmidt GJ, Nicaretta D, van Duinkerken E, Schmidt JJ. Clinical Utility of a Reaction-Time Attention Task in the Evaluation of Cognitive Impairment in Elderly with High Educational Disparity. J Alzheimers Dis. 2021;81(2):691-697. doi: 10.3233/JAD-210151. PMID: 33814451.
- Schmidt GJ, Boechat YEM, van Duinkerken E, Schmidt JJ, Moreira TB, Nicaretta DH, Schmidt SL. Detection of Cognitive Dysfunction in Elderly with a Low Educational Level Using a Reaction-Time Attention Task. J Alzheimers Dis. 2020;78(3):1197-1205. doi: 10.3233/JAD-200881. PMID: 33136095.
- Schumacher R, Halai AD, Lambon Ralph MA. Attention to attention in aphasia - elucidating impairment patterns, modality differences and neural correlates. Neuropsychologia. 2022 Dec 15;177:108413. doi: 10.1016/j.neuropsychologia.2022.108413. Epub 2022 Nov 3. PMID: 36336090; PMCID: PMC7614452.
Additionally, the limitations of the attention outcomes assessed and the related implications for future research were further highlighted in the Discussion section (please see the Discussion section, lines 489-498):
“A significant limitation is the premise that the term' attention' is not a unitary construct, as it encompasses a subset of processes and mechanisms that require further study, especially to provide a single, unified conceptualization [100]. However, based on our findings and supporting evidence from the literature, we believe that we have examined the attention domains most relevant to the outcomes of interest, especially within the context of the neurorehabilitation approach under study in a highly applicable manner to clinical practice [101–103]. Nevertheless, future scientific investigations employing multidimensional neuropsychological tools capable of broader analysis and quantitative detection of attention subsets may expand upon and further delineate our exploratory findings. “
References:
- Hommel, B.; Chapman, C.S.; Cisek, P.; Neyedli, H.F.; Song, J.-H.; Welsh, T.N. No One Knows What Attention Is. Atten Percept Psychophys 2019, 81, 2288–2303, doi:10.3758/s13414-019-01846-w.
- P, M.; As, K.; Gr, F.; S, V. Lateralization, Functional Specialization, and Dysfunction of Attentional Networks. Cortex; a journal devoted to the study of the nervous system and behavior 2020, 132, doi:10.1016/j.cortex.2020.08.022.
- Barker-Collo, S.; Feigin, V.; Lawes, C.; Senior, H.; Parag, V. Natural History of Attention Deficits and Their Influence on Functional Recovery from Acute Stages to 6 Months after Stroke. Neuroepidemiology 2010, 35, 255–262, doi:10.1159/000319894.
- Rj, S.; T, M.; Dk, W.; Je, C.; Jm, R. Poststroke Cognitive Impairment Negatively Impacts Activity and Participation Outcomes: A Systematic Review and Meta-Analysis. Stroke 2021, 52, doi:10.1161/STROKEAHA.120.032215.
The self-developed 8-item AOT engagement questionnaire is an interesting tool, but its validation status remains unclear.
We agree with the Reviewer on the importance of this point. In the Methods section, we have outlined the process of developing the questionnaire and its exploratory approach (please see lines 175-177):
“Items were generated based on a review of engagement constructs in rehabilitation and refined through discussions with the clinical research team.”
However, agreeing with the Reviewer about the limited interpretability of the tool without scientific validation, we have listed the use of an unvalidated questionnaire as a limitation of the manuscript (please see the Discussion section, lines 530-535):
“Another limitation concerns the AOT engagement questionnaire, which was specifically developed for this study. Although designed to capture satisfaction, fatigue, and motivation, the instrument has not undergone formal psychometric validation, which restricts the interpretability of the engagement results. Future research should address this gap by evaluating the reliability and validity of the scale in larger and more diverse cohorts.”
The discussion highlights attentional deficits as predictors of reduced engagement and motor gains, which is well supported. However, alternative explanations (e.g., mood disturbances, cognitive reserve differences) deserve more nuanced integration, since they co-varied with attentional performance.
We thank the Reviewer for this important observation, and we agree. Therefore, we have extensively revised the Discussion section to explicitly acknowledge this limitation and to emphasize the need for future studies with larger samples that allow for statistical control of these variables (please see the revised version of the Discussion). Once again, we thank the Reviewer for giving us the opportunity to enhance the scientific accuracy of our manuscript.
Minor Comments
Abstract: Consider reporting exact subgroup sizes (AD = 6, No-AD = 4) to provide context for readers.
We thank the Reviewer for highlighting this aspect and apologize for not including this key information in the previous version. In response to the Reviewer's comment and previous suggestion, we have revised the abstract as follows (please see the Abstract, lines 22-25):
“At baseline, they were divided into two subgroups based on attentional performance, as determined by scores on the Test of Attentional Performance (subtests of divided attention and Go/No-Go): those with attention deficits (AD, i.e., deficits in one or both tasks, n=6) and those without (No_AD, no deficits in either task, n=4).”
Methods: Please clarify whether therapists were blinded to patients’ attentional classification when delivering AOT. Lack of blinding could bias patient-therapist interaction.
We thank the Reviewer for raising this significant methodological point. We confirm that the therapists delivering AOT were blinded to the patients’ attentional classification throughout the study. This procedure was adopted specifically to minimize the risk of bias in patient–therapist interactions and to ensure that the intervention was provided consistently across all participants. We have now clarified this detail in the Methods section (please see subsection 2.2 AOT protocol, lines 219-221):
“Importantly, therapists delivering AOT were blinded to the patients’ attentional classification to prevent potential bias in patient–therapist interactions and to ensure consistent administration of the intervention.”
Figures: Figures 2–4 are informative but could be improved by including sample sizes (N=) in the legends and ensuring all abbreviations are defined within each caption.
Tables: In Table 2, reporting effect sizes (e.g., Cohen’s d) in addition to p-values would strengthen interpretation, given the small sample.
Language: Minor grammatical inconsistencies (e.g., “patients with attention deficits reported a minor sense of accomplishment” →“patients…reported a reduced sense of accomplishment”).
Comments on the Quality of English Language
Minor grammatical inconsistencies (e.g., “patients with attention deficits reported a minor sense of accomplishment” → “patients…reported a reduced sense of accomplishment”).
We thank the Reviewer for the valuable suggestions provided; we revised the manuscript (including tables, figures, and text) accordingly. Specifically, we have aimed to enhance the quality of the Tables and Figures by implementing the received suggestions. We also had our manuscript reviewed by a native speaker expert to enhance terminological accuracy and the fluency of the discourse. Regarding the effect size, please note, as discussed with another Reviewer, that we believe such a small sample does not allow for obtaining reliable values, and we therefore considered it appropriate not to calculate it. For further information regarding the unreliability of effect size in small samples, please see:
- Button KS, Ioannidis JP, Mokrysz C, et al. Power failure: why small sample size undermines the reliability of neuroscience. Nat Rev Neurosci. 2013;14(5):365-376. doi:10.1038/nrn3475
Reviewer 5 Report
Comments and Suggestions for Authors
General comments:
This manuscript investigates the impact of attentional deficits on patient engagement and motor recovery of upper limb following Action Observation Training (AOT) in subacute stroke patients. Throughout the manuscript, a variety of data analyses are presented, and the choice of terminology is clear and effective. In particular, the Discussion section is well-structured and highly logical, with appropriate references to previous studies that provide strong theoretical support. However, several reporting issues limit the clarity and strength of the findings.
- While the classification of participants into the attention deficit (AD) and No_AD groups is briefly mentioned in the Data Analysis subsection of the Methods section, the specific criteria for this grouping remain insufficiently detailed. Therefore, it is recommended that the authors clearly specify the classification procedure, including the referenced criteria and cutoff scores used to distinguish between the AD and No_AD groups.
- According to the manuscript, the engagement questionnaire was a customized, self-reported scale developed for this study. Even though it is a relatively simple tool applied in the context of a pilot study, some reference to its validity and reliability would strengthen the methodological rigor. Alternatively, it is advisable for the authors to explicitly acknowledge the lack of psychometric validation as a limitation.
- In the Introduction section, it may improve the clarity and structure if the first paragraph is divided before the sentence starting with “Specifically, attentional…”. In this way, the first paragraph would naturally develop from stroke to cognitive issues and then to attentional deficits, while the new second paragraph could focus more specifically on attentional deficits, serving as a typical “problem statement” paragraph.
- In the third sentence of the Data Analysis subsection of the Methods section, the acronym “TAP” is introduced without including the term “test” for the letter “T.” It is recommended that the authors provide the complete definition of the acronym.
- In Table 1b, the neuropsychological and psychological clinical data are clearly presented; however, the large number of measures makes the table appear somewhat complex. It is recommended that the authors explicitly include higher-level categories (e.g., TAP) for each set of measures, which would improve the structural clarity and reader-friendliness of the table.
- Throughout the Results section, there appears to be inconsistent use of the hyphen and en dash. It is recommended that the authors carefully review and standardize their usage for consistency.
- In the third paragraph of Results section, the expression “The improvement in accuracy...” may appears ambiguous and potentially misleading. While it is correct that the No_AD group showed higher accuracy than the AD group at each treatment week, the term “improvement” is misleading, as both groups actually exhibited lower accuracy between W3 and W4. It would be clearer to rephrase this sentence to reflect the consistent between-group difference rather than an improvement over time.
- In the first paragraph of the Results section, it seems that some of the reported partial eta squared (η²ₚ) values appear implausibly large. Verification and correction are recommended.
- In Table 2, the label “Median” may be incorrect. The manuscript does not otherwise report medians, and the median is not conventionally presented with a standard deviation. Please verify whether this is a typographical error and revise the caption/values accordingly.
- For the regression analyses, there appear to be several reporting errors and inconsistencies between the Data Analysis subsection of the Methods section and the Results
- In the Data Analysis section, it is described that variables significantly correlated with the FMA-UE total increment (e.g., mean engagement score) were treated as dependent variables in a regression model. However, this part is introduced as “a logistic regression analysis,” whereas it likely refers to a linear regression model. This seems to be a mislabeling, likely due to a reversal in wording order with the subsequent phrase “a linear regression model.” In addition, the placement of “(dependent variables)” could be misinterpreted as modifying FMA-UE total increment, which may cause confusion. Clarification and correction are strongly recommended.
- In the Introduction section, it may improve the clarity and structure if the first paragraph is divided before the sentence starting with “Specifically, attentional…”. In this way, the first paragraph would naturally develop from stroke to cognitive issues and then to attentional deficits, while the new second paragraph could focus more specifically on attentional deficits, serving as a typical “problem statement” paragraph.
Author Response
This manuscript investigates the impact of attentional deficits on patient engagement and motor recovery of upper limb following Action Observation Training (AOT) in subacute stroke patients. Throughout the manuscript, a variety of data analyses are presented, and the choice of terminology is clear and effective. In particular, the Discussion section is well-structured and highly logical, with appropriate references to previous studies that provide strong theoretical support. However, several reporting issues limit the clarity and strength of the findings.
While the classification of participants into the attention deficit (AD) and No_AD groups is briefly mentioned in the Data Analysis subsection of the Methods section, the specific criteria for this grouping remain insufficiently detailed. Therefore, it is recommended that the authors clearly specify the classification procedure, including the referenced criteria and cutoff scores used to distinguish between the AD and No_AD groups.
We thank the Reviewer for highlighting this important point, and we entirely agree on the importance of clarifying it. As outlined in the Methods section, patients were categorized into groups with or without attentional deficits based on their performance on two subtests of the Test of Attentional Performance (TAP), specifically divided attention and Go/No-Go. Specifically, patients were classified as “without deficits” only if their performance in both subtests fell within the normal range, while an impairment in even one of the two subtests led to classification in the “with attentional deficits” group. This approach was chosen to increase the accuracy of classification for patients without deficits, as previously suggested by other studies. To clarify this important point, we added the following to our manuscript (please see subsection 2.1, lines 156-162):
“Consistent with the literature, impairment on the TAP divided attention and Go/No-Go subtests was defined as performance below the 5th percentile of age- and education-adjusted normative data (collected by the Neuropsychology Service of the Medical Rehabilitation Unit of the Ferrara University Hospital to ensure alignment with the socio-cultural context of the study population), based on reaction times and error rates (omissions and commissions) [47–49]. Due to the specific focus of this work, analyses were conducted on reaction times.”
We hope this clarification addresses the Reviewer’s concern and, at the same time, improves the overall quality and transparency of our work. We are sincerely grateful to the Reviewer for pointing out this important aspect.
References:
- Schmidt SL, Boechat YEM, Schmidt GJ, Nicaretta D, van Duinkerken E, Schmidt JJ. Clinical Utility of a Reaction-Time Attention Task in the Evaluation of Cognitive Impairment in Elderly with High Educational Disparity. J Alzheimers Dis. 2021;81(2):691-697. doi: 10.3233/JAD-210151. PMID: 33814451.
- Schmidt GJ, Boechat YEM, van Duinkerken E, Schmidt JJ, Moreira TB, Nicaretta DH, Schmidt SL. Detection of Cognitive Dysfunction in Elderly with a Low Educational Level Using a Reaction-Time Attention Task. J Alzheimers Dis. 2020;78(3):1197-1205. doi: 10.3233/JAD-200881. PMID: 33136095.
- Schumacher R, Halai AD, Lambon Ralph MA. Attention to attention in aphasia - elucidating impairment patterns, modality differences and neural correlates. Neuropsychologia. 2022 Dec 15;177:108413. doi: 10.1016/j.neuropsychologia.2022.108413. Epub 2022 Nov 3. PMID: 36336090; PMCID: PMC7614452.
According to the manuscript, the engagement questionnaire was a customized, self-reported scale developed for this study. Even though it is a relatively simple tool applied in the context of a pilot study, some reference to its validity and reliability would strengthen the methodological rigor. Alternatively, it is advisable for the authors to explicitly acknowledge the lack of psychometric validation as a limitation.
We agree with the Reviewer on the importance of this point. In the Methods section, we have outlined the process of developing the questionnaire and its exploratory approach (please see lines 175-177):
“Items were generated based on a review of engagement constructs in rehabilitation and refined through discussions with the clinical research team.”
However, agreeing with the Reviewer about the limited interpretability of the tool without scientific validation, we have listed the use of an unvalidated questionnaire as a limitation of the manuscript (please see the Discussion section, lines 530-535):
“Another limitation concerns the AOT engagement questionnaire, which was specifically developed for this study. Although designed to capture satisfaction, fatigue, and motivation, the instrument has not undergone formal psychometric validation, which restricts the interpretability of the engagement results. Future research should address this gap by evaluating the reliability and validity of the scale in larger and more diverse cohorts.”
In the Introduction section, it may improve the clarity and structure if the first paragraph is divided before the sentence starting with “Specifically, attentional…”. In this way, the first paragraph would naturally develop from stroke to cognitive issues and then to attentional deficits, while the new second paragraph could focus more specifically on attentional deficits, serving as a typical “problem statement” paragraph.
We thank the Reviewer for this valuable suggestion, which has indeed helped us achieve a smoother narrative flow and a more coherent logical structure. In response to this recommendation and the constructive input provided by the other Reviewers, we have extensively revised the Introduction (please see the revised version of the Introduction). This section aims to clarify the rationale, more effectively highlight gaps in the literature, and explicitly position our study within the broader research landscape. We trust that these changes adequately address the concerns raised, but we remain fully open to further revising the structure should this be deemed necessary. Once again, we are grateful to the Reviewer for this thorough and insightful evaluation.
In the third sentence of the Data Analysis subsection of the Methods section, the acronym “TAP” is introduced without including the term “test” for the letter “T.” It is recommended that the authors provide the complete definition of the acronym.
We thank the Reviewer for this thoughtful observation. Please note that we have specified the full name of the Test of Attentional Performance (TAP) in the previous Section 2.1 (Study design, lines 143-144). We completely agree on the importance of maintaining consistency in the use of abbreviations and acronyms throughout the manuscript, and we have revised the previous sentence accordingly (please see Section 2.3 Data analysis, line 225):
“Regarding the TAP, all reaction time data are shown as the mean and SD of the raw scores.”
Furthermore, we ensured that this abbreviation, along with all others, is explicitly listed in the Abbreviations section at the conclusion of the manuscript to enhance the reader’s comprehension. Once again, we are grateful to the Reviewer for this valuable suggestion.
In Table 1b, the neuropsychological and psychological clinical data are clearly presented; however, the large number of measures makes the table appear somewhat complex. It is recommended that the authors explicitly include higher-level categories (e.g., TAP) for each set of measures, which would improve the structural clarity and reader-friendliness of the table.
We appreciate the Reviewer for this valuable observation, which has helped us clarify our results better. Specifically, we have improved the legend, added references to the TAP sections, separating them from the clinical-neuropsychological sections and the CRIq (please see the revised version of the Table 1b). We thank the Reviewer again for this helpful suggestion .
Throughout the Results section, there appears to be inconsistent use of the hyphen and en dash. It is recommended that the authors carefully review and standardize their usage for consistency.
We thank the Reviewer for this specification, and we agree. Thus, we have corrected the use of hyphens and en dashes (please see the revised version of the Results).
In the third paragraph of Results section, the expression “The improvement in accuracy...” may appears ambiguous and potentially misleading. While it is correct that the No_AD group showed higher accuracy than the AD group at each treatment week, the term “improvement” is misleading, as both groups actually exhibited lower accuracy between W3 and W4. It would be clearer to rephrase this sentence to reflect the consistent between-group difference rather than an improvement over time.
We thank the Reviewer for this specification, and we agree. Therefore, we carefully checked our Results section and rephrased our sentence to clarify this point (please see the revised version of the Results section, particularly lines 283-296).
In the first paragraph of the Results section, it seems that some of the reported partial eta squared (η²ₚ) values appear implausibly large. Verification and correction are recommended.
We thank the Reviewer for pointing out this error, and we apologise for this inaccuracy. We corrected the sentence with the correct ηp² (please see the revised version of the Results section, particularly lines 283-296).
In Table 2, the label “Median” may be incorrect. The manuscript does not otherwise report medians, and the median is not conventionally presented with a standard deviation. Please verify whether this is a typographical error and revise the caption/values accordingly.
We sincerely thank the Reviewer for pointing this out, and we apologize once again for this inaccuracy. We corrected the label of Table 2 (please see the revised version of Table 2).
For the regression analyses, there appear to be several reporting errors and inconsistencies between the Data Analysis subsection of the Methods section and the Results In the Data Analysis section, it is described that variables significantly correlated with the FMA-UE total increment (e.g., mean engagement score) were treated as dependent variables in a regression model. However, this part is introduced as “a logistic regression analysis,” whereas it likely refers to a linear regression model. This seems to be a mislabeling, likely due to a reversal in wording order with the subsequent phrase “a linear regression model.” In addition, the placement of “(dependent variables)” could be misinterpreted as modifying FMA-UE total increment, which may cause confusion. Clarification and correction are strongly recommended.
We sincerely thank the Reviewer for pointing out this important issue and apologize for the inaccuracies and inconsistencies in reporting the regression analyses. We fully concur with the Reviewer regarding the significance of presenting these aspects with utmost clarity and precision to guarantee both scientific rigor and accurate comprehension by the reader. However, please note that, in line with the other Reviewers' suggestions, we have removed the regression analyses from the revised version of the manuscript. In fact, we endeavored to utilize these analyses to investigate the complex relationship between the variables of interest in the most appropriate manner, to propose potential evidence for future research. Nonetheless, considering the sample size and the exploratory nature of the study, such analyses are inappropriate and the results may be misleading. Therefore, we have removed references to regressions throughout the entire manuscript (please see the revised version). We appreciate the Reviewer for helping us improve the scientific accuracy of our work.
In the Results section, attention deficit is reported as a significant predictor of engagement (binary outcome). By contrast, the description in the Data Analysis section suggests that both patient engagement and mean accuracy were treated as dependent variables in a logistic regression model, which would not be statistically appropriate, as logistic regression requires a single binary dependent variable. Furthermore, in the Results section, the expression “… predictor of the engagement score” would be clearer if reported simply as “predictor of engagement,” in order to reflect its categorical nature.
We thank the Reviewer for this valuable suggestion. However, as explained in our previous response to the Reviewer’s comment, we have removed the regression analysis throughout the entire manuscript due to the exploratory nature of the study and the small sample size. We appreciate the Reviewer's input once again for helping us improve the scientific quality of our work.
Round 2
Reviewer 1 Report
Comments and Suggestions for Authors
The changes to the text made the manuscript suitable for publication. Congratulations to the researchers for their dedication and the work they have completed.
Author Response
The changes to the text made the manuscript suitable for publication. Congratulations to the researchers for their dedication and the work they have completed.
We sincerely thank the Reviewer for taking the time to carefully evaluate the revised version of our manuscript and for providing such positive feedback. We are pleased to have improved the scientific quality and clarity of our work, thanks to the valuable suggestions we received. We would like to thank the Reviewer once again for this opportunity.
Reviewer 2 Report
Comments and Suggestions for Authors
Congratulations to the author. The author has perfectly addressed my concerns. I have no additional suggestions.
Author Response
Congratulations to the author. The author has perfectly addressed my concerns. I have no additional suggestions.
We sincerely thank the Reviewer for taking the time to carefully evaluate the revised version of our manuscript and for providing such positive feedback. We are pleased to have improved the scientific quality and clarity of our work, thanks to the valuable suggestions we received. We would like to thank the Reviewer once again for this opportunity.
Reviewer 3 Report
Comments and Suggestions for Authors
1. Overuse of regression analyses with a very small sample; inconsistency with the authors’ response
Although the authors stated that all regression analyses were removed due to the exploratory nature and small sample size, the revised manuscript still reports adjusted linear and multiple regression models (e.g., F(3)=3.21; R²=.616; p=.048; predictors including attention deficit, stroke severity, and affective symptoms) and labels Figure 4 as a “Linear regression analysis.” These elements remain in the Results and figure captions and are inconsistent with the authors’ response.
Author Response
Overuse of regression analyses with a very small sample; inconsistency with the authors’ response
Although the authors stated that all regression analyses were removed due to the exploratory nature and small sample size, the revised manuscript still reports adjusted linear and multiple regression models (e.g., F(3)=3.21; R²=.616; p=.048; predictors including attention deficit, stroke severity, and affective symptoms) and labels Figure 4 as a “Linear regression analysis.” These elements remain in the Results and figure captions and are inconsistent with the authors’ response.
We sincerely thank the Reviewer for taking the time to evaluate the revised version of our manuscript and for providing this valuable feedback. Please note that, following the Reviewer’s suggestion and the feedback received from the other Reviewers during the first round of revision, and fully agreeing on the importance of clarifying this aspect, we have already removed all references to regression analyses included in the initial submission (please see the revised version of the manuscript). For completeness, we also include below the response previously provided in the first round and approved by the other Reviewers at this stage:
“We thank the Reviewer for highlighting this significant point, and we agree. In fact, we endeavored to utilize these analyses to investigate the complex relationship between the variables of interest in the most appropriate manner, with the aim of proposing potential evidence for future research. Nonetheless, we agree that, considering the sample size and the exploratory nature of the study, such analyses are inappropriate and the results may be misleading. Therefore, also following the suggestion from other Reviewers, we have removed references to regressions throughout the entire manuscript (please see the revised version). We appreciate the Reviewer for helping us improve the scientific accuracy of our work.”
We have further verified and confirm that there are no references to regression analyses in the new version and that Figure 4 (with its caption) has been removed. We appreciate the Reviewer’s feedback, which has further enhanced the scientific quality and transparency of our work.
Reviewer 4 Report
Comments and Suggestions for Authors
The authors have thoughtfully addressed the reviewers’ feedback, making precise revisions that greatly improve the manuscript’s structure and coherence. The technical sections, in particular, are now articulated with clarity, enhancing readability and accessibility for a wider audience. These revisions demonstrate the authors’ rigor and strengthen the manuscript’s overall scholarly contribution, resulting in a polished and impactful work.
Author Response
The authors have thoughtfully addressed the reviewers’ feedback, making precise revisions that greatly improve the manuscript’s structure and coherence. The technical sections, in particular, are now articulated with clarity, enhancing readability and accessibility for a wider audience. These revisions demonstrate the authors’ rigor and strengthen the manuscript’s overall scholarly contribution, resulting in a polished and impactful work.
We sincerely thank the Reviewer for taking the time to carefully evaluate the revised version of our manuscript and for providing such positive feedback. We are pleased to have improved the scientific quality and clarity of our work, thanks to the valuable suggestions we received. We would like to thank the Reviewer once again for this opportunity.